# Relating local connectivity and global dynamics in recurrent excitatory-inhibitory networks

**Yuxiu Shao** *, **Srdjan Ostojic** *

Laboratoire de Neurosciences Cognitives et Computationnelles, INSERM U960, Ecole Normale Superieure—PSL Research University, Paris, France

* yuxiu.shao@ens.psl.eu (YS); srdjan.ostojic@ens.fr (SO)

## Abstract

How the connectivity of cortical networks determines the neural dynamics and the resulting computations is one of the key questions in neuroscience. Previous works have pursued two complementary approaches to quantify the structure in connectivity. One approach starts from the perspective of biological experiments where only the local statistics of connectivity motifs between small groups of neurons are accessible. Another approach is based instead on the perspective of artificial neural networks where the global connectivity matrix is known, and in particular its low-rank structure can be used to determine the resulting low-dimensional dynamics. A direct relationship between these two approaches is however currently missing. Specifically, it remains to be clarified how local connectivity statistics and the global low-rank connectivity structure are inter-related and shape the low-dimensional activity. To bridge this gap, here we develop a method for mapping local connectivity statistics onto an approximate global low-rank structure. Our method rests on approximating the global connectivity matrix using dominant eigenvectors, which we compute using perturbation theory for random matrices. We demonstrate that multi-population networks defined from local connectivity statistics for which the central limit theorem holds can be approximated by low-rank connectivity with Gaussian-mixture statistics. We specifically apply this method to excitatory-inhibitory networks with reciprocal motifs, and show that it yields reliable predictions for both the low-dimensional dynamics, and statistics of population activity. Importantly, it analytically accounts for the activity heterogeneity of individual neurons in specific realizations of local connectivity. Altogether, our approach allows us to disentangle the effects of mean connectivity and reciprocal motifs on the global recurrent feedback, and provides an intuitive picture of how local connectivity shapes global network dynamics.

## Author summary

The structure of connections between neurons is believed to determine how cortical networks control behaviour. Current experimental methods typically measure connections between small numbers of simultaneously recorded neurons, and thereby provide

**Data Availability Statement:** Code can be accessed at https://github.com/shaonannan/ Relating_Local_Conn_Global_Dyns.

**Funding:** YS and SO were supported by the Eranet-Neuron project IMBALANCE and the program

"Ecoles Universitaires de Recherche" launched by the French Government and implemented by the ANR, with the reference ANR-17-EURE-0017. The funders had no role in study design, data collection and analysis, decision to publish, or preparation of the manuscript.

**Competing interests:** There are no competing interests.

information on statistics of local connectivity motifs. Collective network dynamics are however determined by network-wide patterns of connections. How these global patterns are related to local connectivity statistics and shape the dynamics is an open question that we address in this study. Starting from networks defined in terms of local statistics, we develop a method for approximating the resulting connectivity by global low-rank patterns. We apply this method to classical excitatory-inhibitory networks and show that it allows us to predict both collective and single-neuron activity. More generally, our approach provides a link between local connectivity statistics and global network dynamics.

## Introduction

One of the central questions in neuroscience is how the connectivity structure of cortical networks determines the collective dynamics of neural activity and their function. Experimental assessments of connectivity are typically based on measurements of synaptic weights between small numbers of neurons recorded simultaneously [1–9]. The most common approach to quantify connectivity therefore focuses on *local statistics*, and starts by characterizing the connection probability between pairs of neurons based on their type, before considering progressively more complex connectivity motifs. Linking these local connectivity statistics to the emerging network dynamics has been an active topic of investigations [10–24]. A second approach, motivated by computational network models instead of experimental measurements [25–29], instead specifies the connectivity in terms of a low-rank structure defined by network-wide patterns of connectivity [30–39]. This global connectivity structure directly determines the low-dimensional dynamics and the resulting computations [30, 31, 33], yet it remains unclear how it is related to local connectivity statistics that can be recorded experimentally. In this study, we aim to bridge this gap, by mapping local connectivity statistics onto a global, low-rank description of connectivity and comparing the resulting dynamics.

Starting from random networks with connectivity defined in terms of local, cell-type dependent statistics, we develop a low-rank approximation based on the dominant eigenmodes of the connectivity matrix. Using perturbation theory, we show that the obtained low-rank connectivity patterns universally obey Gaussian-mixture statistics and therefore lead to analytically tractable dynamics [31, 33]. We specifically apply this approach to excitatory-inhibitory networks with connections consisting of independent and reciprocal parts, and exploit the low-rank approximation to predict the emerging dynamics.

We first show that, although the dominant low-rank structure is set on average by the mean synaptic weights [40–42], a perturbative approach accurately accounts for the components of individual neurons on the dominant eigenvectors arising from individual instances of the random connectivity. As a result, our low-rank approximation analytically captures the activity of individual neurons in the original E-I network defined based on local statistics. The analytic description of the dynamics in the low-rank approximation moreover leads to the identification of two distinct sources of recurrent feedback corresponding respectively to the mean connectivity and reciprocal connections between neurons. In particular, the reciprocal motifs impact dynamics by modulating both the dominant eigenvalue and the corresponding eigenvectors, and can give rise to additional bistability in the network. Altogether, our analytical mapping of the local EI statistics to a low-rank description provides a quantitative and intuitive description of how local connectivity statistics determine global low-dimensional dynamics.

**Table 1. List of notations.**

| Notation | Description |
|---|---|
| $i, j$ | Single neuron indices |
| $p, q$ | Neuron population indices |
| $\alpha_p$ | Fraction of neurons belonging to population $p$ |
| $\bar{J}_{pq}$ | Mean synaptic weights between populations $p, q$ |
| $J_{E/I}$ | Re-scaled mean excitatory/inhibitory synaptic weights |
| $\sigma_{z_{pq}}$ | Standard deviation of the synaptic weights between populations $p, q$ |
| $g_{pq}$ | Re-scaled standard deviation of the synaptic weights between populations $p, q$ |
| $\eta_{pq}$ | Reciprocal correlation of connectivity weights between populations $p, q$ |
| $A_{E/I}$ | Excitatory/inhibitory synaptic weights in the sparse network |
| $c$ | Homogeneous sparsity of the sparse network, $c_{pq} = c$ |
| $\sigma_{m^p}^2, \; \sigma_{n^p}^2$ | Variances of components on connectivity vectors $\mathbf{m}^p$ and $\mathbf{n}^p$ |
| $\sigma_{nm}^p$ | Covariance between connectivity vectors $\mathbf{m}^p$ and $\mathbf{n}^p$ |
| $\mu_x^p, \; \Delta_x^p$ | Population-averaged mean and variance of activation |

The paper is structured as follows. In Results Secs. 1.1, 1.2 we explain our setup for the local and global representations of random connectivity, and introduce the low-rank approximation, matrix determinant lemma and perturbation theory which are the basis of our analytical approach. We first start from the case of Gaussian excitatory-and-inhibitory networks, Results Secs. 1.3, 1.4 demonstrate our analysis for approximating local connectivity with low-rank structure. In the following Results Sec. 1.5, we use the low-rank approximation models obtained to describe the nonlinear dynamics. Then, in Results Sec. 1.6 we generalize our analysis to sparse excitatory-and-inhibitory networks. Major novel findings, key insights, restrictions and future extensions are addressed in discussions. Technical details and a retour to the Gaussian-mixture low-rank framework are covered in methods, preceded by a table of notations (Table 1).

# 1 Results

## 1.1 Local vs. global representations of random recurrent connectivity

We study networks of $N$ rate units with random recurrent connectivity given by the connectivity matrix $\mathbf{J}$, where the entry $J_{ij}$ corresponds to the strength of the synapse from neuron $j$ to neuron $i$. A full statistical description of the random connectivity would require specifying the joint distribution $P(\{J_{ij}\})$ of the $N^2$ synaptic weights. Determining the dynamics from this high-dimensional distribution is however in general intractable. We therefore focus on connectivity models that make simplifying assumptions on the underlying statistics.

Our specific goal is to relate two different classes of such models, which we refer to as the *local* and the *global* representations of recurrent connectivity. Both representations assume that the network consists of $P$ populations, and the statistics of connectivity depend only on the pre- and post-synaptic populations. The two representations are however based on different statistical features of the connectivity.

The local representation defines the connectivity statistics by starting from the marginal distributions Prob($J_{ij} = J$) of individual synaptic weights, and by including progressively higher-order correlations referred to as *connectivity motifs* [1, 10, 14]. In this work, we will consider only the first two orders, i. e. the distribution of individual weights and the pairwise correlations $\eta_{ij}$ between reciprocal connections $J_{ij}$ and $J_{ji}$ that quantify pairwise motifs

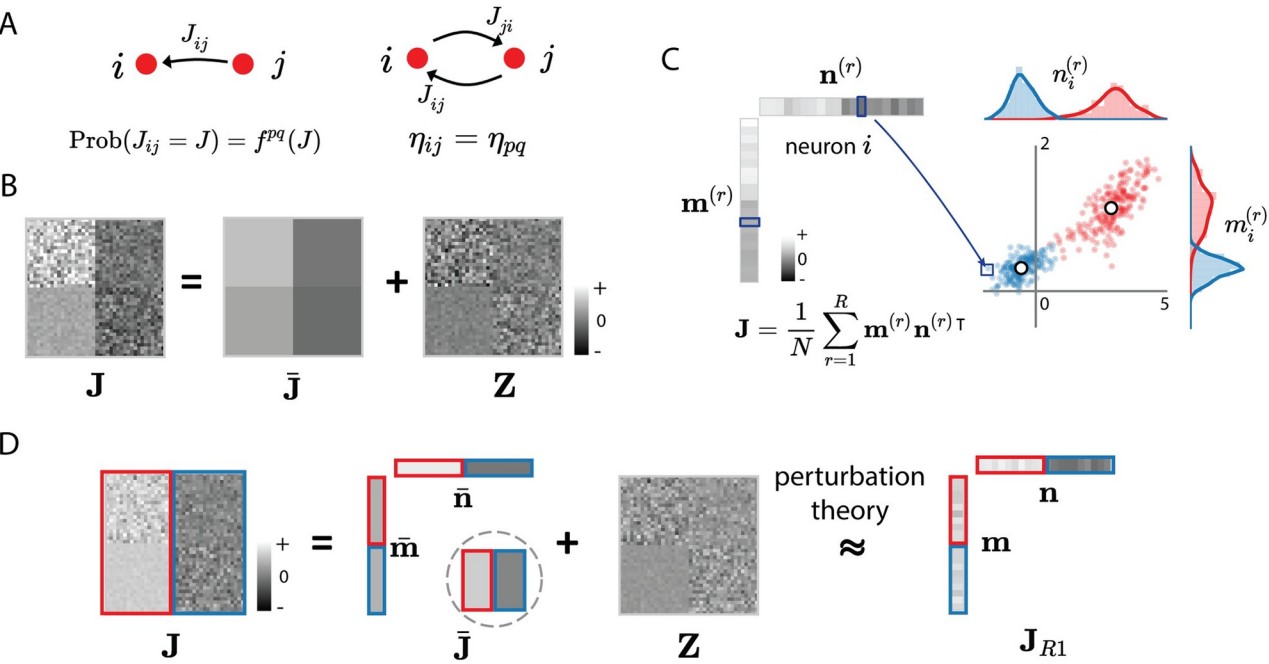

**Fig 1. Local vs global representations of recurrent connectivity.** (A) The local representation defines the statistics of synaptic weights $J_{ij}$ by starting from the marginal probability distribution of individual synaptic weights (left) and then specifying reciprocal motifs in terms of correlations $\eta_{ij}$ between reciprocal weights $J_{ij}$ and $J_{ji}$ connecting neurons $i$ and $j$ (right). Both the marginal distribution and the reciprocal correlations are assumed to depend only on the populations $p$ and $q$ that the neurons $i$ and $j$ belong to. (B) The resulting connectivity matrix $\mathbf{J}$ has block-structured statistics, where different blocks correspond to connections between the $P$ different populations ($P = 2$ in this illustration). It can be decomposed into a superposition of a mean component $\bar{\mathbf{J}}$, and a remaining zero-mean random connectivity component $\mathbf{Z}$ that has block-structured variances. (C) The global, low-rank representation defines the connectivity matrix $\mathbf{J}$ as the sum of $R$ outer products between connectivity vectors $\mathbf{m}^{(r)}$, $\mathbf{n}^{(r)}$ for $r = 1 \ldots R$. The statistics of connectivity are defined in terms of the joint probability distribution over neurons $i$ of their entries $(m_i^{(1)}, \ldots, m_i^{(R)}, n_i^{(1)}, \ldots, n_i^{(R)})$ on connectivity vectors. We specifically consider the class of Gaussian-mixture low-rank models, where each neuron is first assigned to a population $p$, and within each population the entries on connectivity vectors are generated from a multivariate Gaussian distribution with fixed statistics. Here we illustrate this distribution for one pair of connectivity vectors ($R = 1$) and $P = 2$ populations. Each dot represents the connectivity parameters $(m_i^{(r)}, n_i^{(r)})$ of one neuron $i$, the red and blue colours denote the two populations, white dots and the rotations of the dot clouds indicate the mean and covariance of the distribution for each population. (D) Relating the local and global representations of recurrent connectivity for a simplified excitatory-inhibitory network. In this model, the mean connectivity depends only on the presynaptic population (indicated by red and blue colours). The mean connectivity $\bar{\mathbf{J}}$ is in this case rank-one, and can be written as an outer product of vectors $\bar{\mathbf{m}}$ and $\bar{\mathbf{n}}$. We approximate the full connectivity by a rank-one matrix, with connectivity vectors $\mathbf{m}$ and $\mathbf{n}$ obtained from $\bar{\mathbf{m}}$ and $\bar{\mathbf{n}}$ using perturbation theory.

(Fig 1A). Our key assumption is that both the marginal distributions of $J_{ij}$ and the correlations $\eta_{ij}$ depend only on the populations $p$ and $q$ that the post- and pre-synaptic neurons belong to:

$$\text{Prob}(J_{ij} = J) = f^{pq}(J), \quad p, q = 1 \ldots P,$$
$$\eta_{ij} = \eta_{pq}. \tag{1}$$

All synapses connecting the same two populations therefore have identical statistics, leading to a block-like statistical structure for the connectivity matrix $\mathbf{J}$ (Fig 1B left panel).

The global representation of connectivity instead refers to the situation where $\mathbf{J}$ is defined as a low-rank matrix [30, 31, 33]:

$$\mathbf{J} = \frac{1}{N} \sum_{r=1}^{R} \mathbf{m}^{(r)} \mathbf{n}^{(r)\mathsf{T}}. \tag{2}$$

Here $\mathbf{m}^{(r)} = \{m_i^{(r)}\}_{i=1...N}$ and $\mathbf{n}^{(r)} = \{n_i^{(r)}\}_{i=1...N}$ for $r = 1...R$ are referred to as *connectivity vectors*, where $R$ is the rank of $\mathbf{J}$. In this representation, the statistics of connectivity are defined by the distribution of vector elements, rather than directly by the distribution of synaptic weights as in the local representation. Specifically, each neuron $i$ is characterized by its set of entries $(m_i^{(1)}, \ldots, m_i^{(R)}, n_i^{(1)}, \ldots, n_i^{(R)})$ over the connectivity vectors. For each neuron, these $2R$ entries are generated from a joint distribution, independently of the other neurons, and the parameters of this joint distribution depend on the population $p$ the neuron belongs to. Here we focus on the broad class of Gaussian-mixture low-rank networks, in which for population $p$, the joint distribution of elements is a multi-variate Gaussian defined by the means and covariances of the $2R$ entries [31, 33] (Fig 1C).

To relate the local and the global representations of connectivity, a key observation is that any matrix $\mathbf{J}$ generated from the local statistics defined in Eq (1) can be expressed as

$$\mathbf{J} = \bar{\mathbf{J}} + \mathbf{Z} \tag{3}$$

where $\bar{\mathbf{J}}$ contains the mean values of the connections, and $\mathbf{Z}$ contains the remaining, zero-mean random part [40]. Because of the underlying population structure (Eq (1)), $\bar{\mathbf{J}}$ consists of $P \times P$ blocks with identical values within each block (Fig 1B middle panel), and is therefore at most of rank $P$. The random part $\mathbf{Z}$ is instead in general of rank $N$, but obeys block-like statistics, with variance and normalized covariance parameters defined by $P \times P$ matrices (Methods Secs. 2.1.1–2.1.2, Eqs (27) and (32)).

For the sake of simplicity, in this study, we focus on a simplified excitatory-inhibitory model [41, 43]. This network consists of one excitatory and one inhibitory population, so $P = 2$ and in the following we use the population indices $p, q = E, I$. A central simplifying assumption in this model is that the mean synaptic weights depend only on the pre-synaptic population, so that $\bar{J}_{EE} = \bar{J}_{IE} > 0$ and $\bar{J}_{II} = \bar{J}_{EI} < 0$. The mean connectivity matrix $\bar{\mathbf{J}}$ therefore consists of only two blocks and is unit rank (Fig 1D). The statistics of the random part $\mathbf{Z}$ instead depend on both pre- and post-synaptic populations, and are therefore described by $2 \times 2$ matrices of variance and normalized covariance parameters (see Methods Sec. 2.1.3, Eq (37)).

## 1.2 Approximating locally-defined connectivity with low-rank connectivity

To relate the local and global representations of connectivity, we start from a connectivity matrix $\mathbf{J}$ generated from the local statistics (Eq (1)) and approximate it by a rank-$R$ matrix of the form given in Eq (2). As the locally-defined connectivity matrix $\mathbf{J}$ is of rank $N$, this is equivalent to the classical low-rank approximation problem, for which a variety of methods exist [30, 31, 33]. Here we use simple truncated eigen-decomposition as it preserves the dominant eigenvalues that determine nonlinear dynamics.

Applying the standard eigenmode decomposition, $\mathbf{J}$ can be in general factored as

$$\mathbf{J} = \frac{1}{N} \sum_{r=1}^{N} \mathbf{m}^{(r)} \mathbf{n}^{(r)\mathsf{T}}, \tag{4}$$

where $\mathbf{m}^{(r)}$ and $\mathbf{n}^{(r)}$ are rescaled versions of the $r$-th right and left eigenvectors (Methods Sec. 2.3, Eqs (44)–(49)), ordered by the absolute value of their eigenvalue $\lambda_r$ for $r = 1...N$. A rank-$R$ approximation that preserves the top $R$ eigenvalues can then be obtained by simply keeping the first $R$ terms in the sum in Eq (4). In this study, we focus on $R = 1$, corresponding to the dominant eigenvalue. Higher rank approximations will be described elsewhere.

Eigenvalues and eigenvectors are in general complex nonlinear functions of the entries of the matrix $\mathbf{J}$. To determine the dominant eigenvalues and the corresponding vectors of $\mathbf{J}$, we

capitalize on the observation in Eq (3) that a locally-defined connectivity matrix can in general be expressed as a sum of a low-rank matrix of mean values $\bar{\mathbf{J}}$ and the remaining random part $\mathbf{Z}$. Previous studies have found that the eigenspectra of matrices with such structure typically consist of two components in the complex plane: a continuously-distributed bulk determined by the random part, and discrete outliers controlled by the low-rank structure [30, 32, 39, 44–46]. In this study, we extend previous approaches to determine the influence of the block-like statistics of $\mathbf{Z}$ on the outliers that correspond to dominant eigenvalues. We then use perturbation theory to determine the corresponding left and right eigenvectors and their statistical structure. Here we summarize the main steps of this analysis (full details are provided in Methods), and then apply it to specific cases in the following sections.

We focus on the simplified E-I network for which the mean part of the connectivity is unit rank and can therefore be written as $\bar{\mathbf{J}} = \bar{\mathbf{m}}\bar{\mathbf{n}}^{\mathsf{T}}/N$, so that the full connectivity matrix is

$$\mathbf{J} = \frac{1}{N}\bar{\mathbf{m}}\bar{\mathbf{n}}^{\mathsf{T}} + \mathbf{Z}. \tag{5}$$

The mean part $\bar{\mathbf{J}}$ of the connectivity has a unique non-trivial eigenvalue $\lambda_0 = \bar{\mathbf{n}}^{\mathsf{T}}\bar{\mathbf{m}}/N$ which can give rise to one or several outliers $\lambda$ in the eigenspectrum of $\mathbf{J}$. To determine how the random part of the connectivity influences $\lambda$, we start from the characteristic equation for the eigenvalues of $\mathbf{J}$ and exploit Eq (5) to apply the matrix determinant lemma (Eq (59)) and get

$$\lambda = \frac{1}{N}\bar{\mathbf{n}}^{\mathsf{T}}(\mathbf{I} - \mathbf{Z}/\lambda)^{-1}\bar{\mathbf{m}}. \tag{6}$$

As long as $(\mathbf{I} - \mathbf{Z}/\lambda)$ is invertible, this equation determines the eigenvalues of $\bar{\mathbf{m}}\bar{\mathbf{n}}^{\mathsf{T}}/N + \mathbf{Z}$.

We next assume that the maximal eigenvalue of $\mathbf{Z}$ is smaller than the outlying eigenvalues corresponding to the low-rank approximation $\mathbf{m}\mathbf{n}^{\mathsf{T}}/N$ we aim to determine (Eq (4), Methods Sec. 2.2 Eqs (40) and (41)). This condition holds as long as the variance amplitudes $g_{pq}^2/N$ of the random part $\mathbf{Z}$ of the connectivity are not too large [15, 18](S4 Text). This assumption allows us to do series expansion which leads to a nonlinear equation for $\lambda$ [32]:

$$\lambda = \sum_{k=0}^{\infty}\frac{\theta_k}{\lambda^k} \quad \text{with} \quad \theta_k = \frac{1}{N}\bar{\mathbf{n}}^{\mathsf{T}}\mathbf{Z}^k\bar{\mathbf{m}}. \tag{7}$$

Although this nonlinear equation is a polynomial with infinite terms, there are at most finite $N$ solutions for the eigenvalue outliers [32]. More specifically, in this work, we are only focusing on the second-order reciprocal motifs in the random component. The second order coefficient $\theta_2$ in Eq (7) is the first non-trivial term for the reciprocal case, so we truncate the series summation at $k = 2$ (included) to provide a more straightforward and understandable comprehension of how reciprocal motifs modify the eigenvalue outlier. In addition, we give the computation of eigenvalue outliers without truncation, and more elaborate explanations in S6 Text.

Truncating the sum to second order yields an approximate third order polynomial for $\lambda$:

$$\lambda^3 = \lambda_0\lambda^2 + \theta_1\lambda + \theta_2, \quad \text{with} \quad \theta_1 = \bar{\mathbf{n}}^{\mathsf{T}}\mathbf{Z}\bar{\mathbf{m}}/N \quad \theta_2 = \bar{\mathbf{n}}^{\mathsf{T}}\mathbf{Z}^2\bar{\mathbf{m}}/N. \tag{8}$$

The statistics of the outlying eigenvalue can then be obtained by averaging over the random part of the connectivity $\mathbf{Z}$.

An approximate expression for the right and left connectivity vectors $\mathbf{m}$ and $\mathbf{n}$ of $\mathbf{J}$ corresponding to the outliers $\lambda$ can be determined using first order perturbation theory [47], where we treat the variance amplitudes $g_{pq}^2/N$ of the random part $\mathbf{Z}$ of the connectivity as small

parameters. We first note that $\bar{\mathbf{m}}$ and $\bar{\mathbf{n}}$ are the right- and left-eigenvectors of $\bar{\mathbf{J}}$ corresponding to the non-trivial eigenvalue $\lambda_0$. Interpreting the full connectivity matrix $\mathbf{J}$ as $\bar{\mathbf{J}}$ perturbed by a random matrix $\mathbf{Z}$, at first order $\mathbf{m}$ and $\mathbf{n}$ can be expressed as

$$
\begin{aligned}
\mathbf{m} &= \bar{\mathbf{m}} + \Delta\mathbf{m} \\
\mathbf{n} &= \bar{\mathbf{n}} + \Delta\mathbf{n},
\end{aligned}
\tag{9}
$$

with

$$
\begin{aligned}
\Delta\mathbf{m} &= \mathbf{Z}\bar{\mathbf{m}}/\lambda_0 \\
\Delta\mathbf{n} &= \mathbf{Z}^{\mathsf{T}}\bar{\mathbf{n}}/\lambda_0.
\end{aligned}
\tag{10}
$$

A key observation is that each element of $\Delta\mathbf{m}$ and $\Delta\mathbf{n}$ is a sum of $N$ random variables. If the elements $\{z_{ij}\}$ in $\mathbf{Z}$ are random samples drawn from a distribution with overall mean and finite variance, the central limit theorem holds and therefore predicts that, in the limit of large $N$, the statistics of $\Delta m_i$ and $\Delta n_i$, and therefore $m_i$ and $n_i$, follow a Gaussian distribution. In general, the mean and variance of $m_i$ and $n_i$ and their correlation are determined by the mean, variance and correlation of the elements of $\mathbf{J}$, but not the specific form of the probability distribution. Since the matrix $\mathbf{Z}$ has block-like statistics determined by the population structure, the statistics of the resulting $m_i$ and $n_i$ depend on the population $p$ the neuron $i$ belongs to. Overall, the distribution of elements of $\mathbf{m}$ and $\mathbf{n}$ obtained from perturbation theory therefore follow a *Gaussian-mixture* distribution, so that our approach effectively approximates a locally-defined $\mathbf{J}$ by a Gaussian-mixture low-rank model specified by the means $\bar{m}^p$, $\bar{n}^p$, the variances $\sigma_{m^p}^2$, $\sigma_{n^p}^2$ and the covariances $\sigma_{nm}^p$ of the entries on the connectivity vectors for $p = E, I$.

We next apply the perturbative approach described here to networks with independent random components, and then to networks with reciprocal motifs.

### 1.3 Low-rank structure induced by independently generated synaptic connections

We first apply our approach for a low-rank approximation to the simplest version of the locally-defined excitatory-inhibitory network where each $J_{ij}$ is generated independently from a Gaussian distribution with a mean that depends only on the pre-synaptic population, i. e. $[J_{ij}] = \bar{J}_{pq} = \bar{J}_q$ with $p, q \in E, I$. The entries of the eigenvectors $\bar{\mathbf{m}}$ and $\bar{\mathbf{n}}$ of the mean connectivity matrix $\bar{\mathbf{J}}$ are then given by:

$$
\bar{m}_i = 1, \quad i = 1 \ldots N
\tag{11}
$$

$$
\bar{n}_i = \bar{n}^E = \frac{N}{N_E}J_E \quad i \in N_E
\tag{12}
$$

$$
\bar{n}_i = \bar{n}^I = -\frac{N}{N_I}J_I \quad i \in N_I.
\tag{13}
$$

In the large network limit, averaging over $\mathbf{Z}$ in Eq (7) yields $[\theta_k] = 0$ for all $k > 0$ [32], so that the outlier is on average given by $[\lambda] = \theta_0 = \lambda_0$ (Fig 2A). Our approach moreover gives an expression for the standard deviation of the outlier in the finite-size network, which grows linearly with $g$ (Fig 2B, Eq (80)). Examining the entries of the left and right eigenvectors $\mathbf{n}$ and $\mathbf{m}$ of $\mathbf{J}$ corresponding to the outlier, as expected, the distribution of $(m_i, n_i)$ is well described by a mixture of two Gaussians centred at $\bar{m}^p$, $\bar{n}^p$ (Fig 2C). We further find that perturbation theory

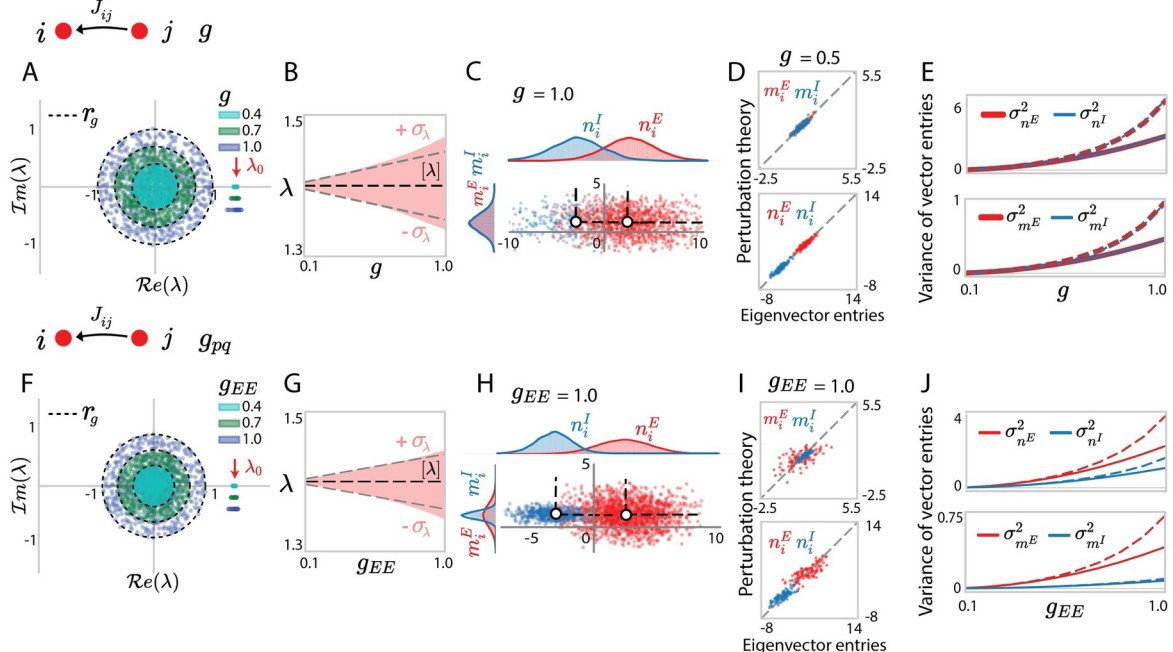

**Fig 2. Eigenvalues and dominant eigenvectors for locally-defined Gaussian connectivity with independent synaptic weights.** (A) Eigenvalue spectra of excitatory-inhibitory connectivity matrices **J** with elements generated from Gaussian distributions with identical variances $g^2/N$ over neurons. The coloured dots in the circular bulk shows 600 eigenvalues for one realization of the random connectivity for each value of $g$. Different colours correspond to different values of $g$. Dashed envelopes indicate the theoretical predictions for the radius $r_g$ of the circular bulk computed according to Eqs. (147), (148). Outlying eigenvalues are shown for 30 realizations of the random connectivity, and for different $g$ their location on the $y$-axis is shifted to help visualization, the dispersion reflects finite-size effects. The red arrow points to the eigenvalue $\lambda_0$ of the mean connectivity matrix **J̄**. (B) Statistics of outlying eigenvalues over realizations of random connectivity. Empirical distribution (the red area shows mean ± standard deviation and reflects finite-size effects), compared with the theoretical predictions for the mean (black dashed line) and standard deviation (gray dashed line) obtained using Eq (80). (C) Scatter plot showing for each neuron $i$ its entry $n_i$ on the left eigenvector against its entry $m_i$ on the right eigenvector. Red and blue colours represent respectively excitatory and inhibitory neurons. The white dots and the dashed lines respectively indicate the means and covariances for each population obtained from simulations. (D) Comparison between eigenvector entries obtained from direct eigen-decomposition of **J** with projections obtained using perturbation theory (Eqs (9) and (10)) in a given realization of **Z**. (E) Comparison between simulations (dashed lines) and theory (full lines) for the variances $\sigma_{n^p}^2$, $\sigma_{m^p}^2$ of eigenvector entries corresponding to different populations (Eq (90)). To help visualization, the curves for the excitatory population are thicker than the curves for the inhibitory population. (F-J) Identical quantities for connectivity matrices in which the variance parameters are heterogeneous: $g_{EE} : g_{EI} : g_{IE} : g_{II} = 1.0 : 0.5 : 0.2 : 0.8$, $g_{EE}$ increases from 0 to 1. Other network parameters $N_E = 4N_I = 1200$ and $J_E = 2.0$, $J_I = 0.6$ in all simulations.

accurately accounts for the individual entries of the eigenvectors as long as $g$ is sufficiently below unity (Fig 2D). Perturbation theory provides a lower bound for the values of the corresponding variances. For large values of $g$, the distributions remain Gaussian, but their variances increase above the predictions of perturbation theory (Fig 2E). The reason for this deviation from the theory is that perturbation theory assumes a small value of the scaling factor $g$ in the variance of $J_{ij}$; as a result, the deviation also reflects the systematic error resulting from the first-order approximation. Importantly, the entries of the left and right eigenvectors are uncorrelated, and only their means, but not their variances, differ between the two populations.

We next turn to the case where the variances of synaptic weights depend on the pre- and post-synaptic populations $q$, $p$, and are given by $g_{pq}^2/N$. In that case, the entries of the random part of the connectivity **Z** are independent, but not identically distributed Gaussians. Previous studies [11, 44] have shown that the spectrum of **Z** remains circularly symmetric, but its radius $r_g$ is determined by a combination of variance parameters $g_{pq}$ (S4 Text, Eqs. (147), (148)).

Examining the resulting connectivity matrix $\mathbf{J}$, we found that the results for the uniform case directly extend to this heterogeneous situation. The eigenspectrum of $\mathbf{J}$ still consists of an independent superposition of the spectra of $\mathbf{Z}$ and $\bar{\mathbf{J}}$ (Fig 2F). In particular, the random part of the connectivity does not modify the average value of the outlier, but only impacts its variance, which now depends on a combination of the variances $g_{pq}$ (Fig 2G, Eq (90)). Similarly to the uniform case, the distribution of the entries of the left and right eigenvectors is well described by a mixture of two Gaussians, with variances predicted by perturbation theory. The entries of the left and right eigenvectors are uncorrelated, but now both their means and variances depend on the population the neuron belongs to (Fig 2H–2J).

In summary, when synaptic connections $J_{ij}$ are generated independently across pairs of neurons, the equivalent global representation is a Gaussian-mixture low-rank model where the entries of the structure vectors are independent with mean values determined by the low-rank structure of the mean connectivity matrix $\bar{\mathbf{J}}$. Although the mean connectivity establishes the network's basic structure, the Gaussian-mixture low-rank approximation improves on it in terms of preserving connectivity fluctuations and reflecting the cell-type-dependent structural variances in the original locally-defined random connectivity part $\mathbf{Z}$. Importantly, in that situation, the dominant outlying eigenvalues of $\mathbf{J}$ are on average identical to those of $\bar{\mathbf{J}}$, that is, $[\lambda] = \lambda_0$.

## 1.4 Low-rank structure induced by reciprocal motifs

We next turn to locally-defined excitatory-inhibitory networks with reciprocal connectivity motifs quantified by the correlation $\eta_{ij}$ between reciprocal synaptic weights $J_{ij}$ and $J_{ji}$. We assumed that these reciprocal correlations are identical for any pair of neurons $i$ and $j$ belonging to a given pair of populations $p$ and $q$, and used the corresponding parameters $\eta_{pq}$ to generate the connectivity matrix $\mathbf{J}$ (Methods Sec. 2.1.2). Within the decomposition of $\mathbf{J}$ in a mean $\bar{\mathbf{J}}$ and random part $\mathbf{Z}$ (Eq (24)), the additional reciprocal correlations affect only the statistics of $\mathbf{Z}$.

We first consider the homogeneous case where the reciprocal correlation is identical across all populations, i. e. $\eta_{pq} = \eta$ (Fig 3). Previous studies have shown that a random matrix $\mathbf{Z}$ with zero mean and reciprocal correlations $\eta$ has a continuous spectrum that is deformed from a circle into an ellipse as $\eta$ is increased [18, 48]. Superpositions between correlated random matrices, and low-rank structure such as $\bar{\mathbf{J}}$ have, to our knowledge, not been previously studied. Inspecting the eigenspectrum of $\mathbf{J} = \bar{\mathbf{J}} + \mathbf{Z}$, we find that it still consists of a continuous bulk and discrete outliers (Fig 3A). The continuous bulk is contained in an ellipse in the complex plane identical to the spectrum of $\mathbf{Z}$, as in the uncorrelated case. In contrast, we find that the outliers deviated from the eigenvalues of $\bar{\mathbf{J}}$ as $\eta$ is increased (Fig 3B). These deviations are well captured by our analytic approach summarized in Eq (8). Indeed, when averaging Eq (8) over $\mathbf{Z}$, reciprocal correlations generate a non-zero $[\theta_2] = [\bar{\mathbf{n}}^{\mathsf{T}} \mathbf{Z}^2 \bar{\mathbf{m}}]/N$ due to $\mathbf{Z}^2$. This term leads to a cubic equation in Eq (8) and therefore has two effects. First, the non-zero $\theta_2$ induces deviations of the outliers from the eigenvalue $\lambda_0$ of $\bar{\mathbf{J}}$. The direction of these deviations is positive if excitation dominates ($\lambda_0 = J_E - J_I > 0$) and negative if inhibition dominates ($\lambda_0 = J_E - J_I < 0$, Fig 3G–3I). Second, the cubic equation can have up to three solutions and therefore potentially generates additional outliers, and in particular complex conjugate ones. Whether these additional outliers are observed depends on the accuracy of the third-order approximation (Eq (8)) to the determinant lemma (Eq (59)), and on the norm of these outliers compared to the spectral radius, in both scenarios where the network has a homogenous variance $g^2/N$ (Fig 3A and 3G) and when $g_{pq}$ differ between populations (Fig 3J). Please refer to S6 Text for a more thorough discussion.

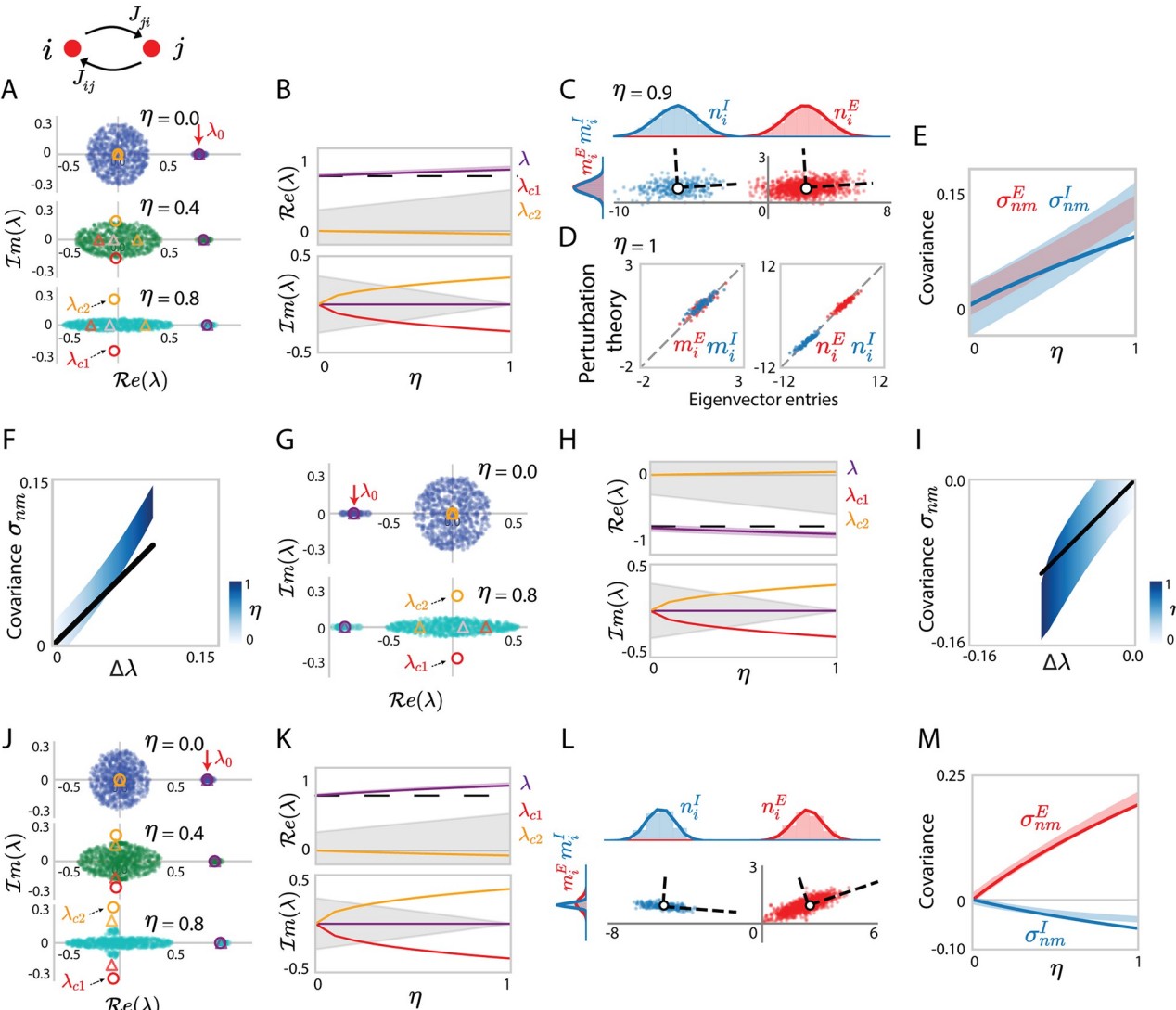

**Fig 3. Eigenvalues and dominant eigenvectors for locally-defined Gaussian connectivity with *homogeneous* reciprocal correlations.** (A) Eigenvalue spectra of excitatory-inhibitory connectivity matrices **J**, with homogeneous reciprocal correlations $\eta$. Different colours from top to bottom correspond to networks with different values of $\eta$. The dots in the elliptical bulk show 600 eigenvalues for one realization of the random connectivity. Outlying eigenvalues are shown for 30 realizations of the random connectivity, the dispersion reflects finite-size effects. The red arrow on the top points to the eigenvalue $\lambda_0$ of the mean connectivity $\bar{\mathbf{J}}$. Coloured circles are the eigenvalues predicted using determinant lemma and truncated series expansion (Eqs (7) and (8), truncating at $k = 2$), coloured triangles are the eigenvalues predicted using determinant lemma without finite truncation (see S6 Text, Eqs. (153)-(157)). (B) Comparison of the eigenvalues from the finite-size simulation with the predictions of the determinant lemma as the reciprocal correlation $\eta$ is increased. The coloured solid lines show the roots of the third-order polynomial in Eq (8) (truncated series expansion). The light purple area indicates the empirical distribution of the dominant outlier for 30 realizations, reflecting finite-size effects, while the black dashed line is the unperturbed eigenvalue $\lambda_0$. The grey areas represent the areas covered by the eigenvalue bulk. (C) Scatter plot showing for each neuron $i$ its entry $n_i$ on the left eigenvector against its entry $m_i$ on the right eigenvector. Red and blue colours represent respectively excitatory and inhibitory neurons. The white dots and the dashed lines respectively indicate the means and covariances for each population. (D) Comparison between eigenvector entries obtained from direct eigen-decomposition of **J** with projections obtained using perturbation theory (Eqs (9) and (10)) in a given realization of **Z**. (E) Comparison between simulations (coloured areas, finite-size effects) and theoretical predictions (coloured lines, Eq (97)) for the population covariance $\sigma_{nm}^p$ of the entries on the left and right connectivity eigenvectors to different populations. (F) Comparison of the overall covariance $\sigma_{nm}$ (Eq (72)) with the deviation $\Delta\lambda$ of the dominant outlying eigenvalue from the unperturbed value $\lambda_0$. Empirical covariance (gradient blue area reflects finite-size effects, where the colour depth represents $\eta$) compared with the theoretical prediction (black line) obtained using Eqs (97) and (92). The $x$-axis uses the theoretical prediction of the deviation of the eigenvalue $\lambda$ from $\lambda_0$. Other network parameters: $J_E = 2.0$, $J_I = 1.2$, $N_E = 4N_I = 1200$ and homogeneous variance parameters $g_{pq} = g = 0.3$. (G-I) Same as (A, B, F) for an inhibition dominates connectivity matrix where $J_I = 2.0$, $J_E = 1.2$, with homogeneous reciprocity $\eta$ and variance parameters $g = 0.3$. (J-M) Same as (A-C) and (E) for excitatory-and-inhibitory connectivity matrices with homogeneous reciprocal correlations $\eta$ but cell-type-dependent variance parameters $g_{EE} : g_{EI} : g_{IE} : g_{II} = 1.0 : 0.5 : 0.2 : 0.8$ and $g_{EE} = 0.3$. In (B, E, F, H, I, K, M), the departure of the centres of the numerical simulation results from the theoretical predictions reflect the systematic errors due to the first-order perturbation approximation of eigenvalues and eigenvectors.

We next examine the right- and left-eigenvectors **m** and **n** corresponding to the dominant outlier. Analogous to the uncorrelated case in the networks with independent connections, perturbation theory accounts for the individual entries of these vectors from a specific realization of the locally-defined random connectivity **Z**, and these individual entries $m_i$ and $n_i$ exhibit Gaussian-mixture statistics (Fig 3C and 3D, Eq (10), and Methods Sec. 2.3 Eqs (69) and (70)). Unlike in the uncorrelated case, reciprocal correlations now induce correlations between $\Delta m_i$ and $\Delta n_i$ (Methods Sec. 2.5.2). Indeed, perturbation theory predicts that the first-order effects $\Delta \mathbf{m}$ and $\Delta \mathbf{n}$ of the random connectivity on **m** and **n** are respectively determined by **Z** and its transpose $\mathbf{Z}^\top$ (Eq (10)). Reciprocal correlations between $z_{ij}$ and $z_{ji}$ directly lead to correlations between **Z** and $\mathbf{Z}^\top$ and therefore a non-zero covariance $\sigma_{nm}$ between elements of **m** and **n**, that can be predicted by mean field theory (Fig 3E, Eq (72)). The strength of the covariance between eigenvector entries reflects the strength of the additional feedback loop due to reciprocal correlations, and is therefore directly related to the deviations of the outlying eigenvalue from the uncorrelated value $\lambda_0$ (Eqs (42) and (96), Fig 3F). When the network has both homogeneous variance parameters and correlation parameters, the excitatory and inhibitory populations have the same covariance $\sigma_{nm}^E = \sigma_{nm}^I = (J_E - J_I)g^2\eta/\lambda^2$ (Eq (97)). If the synaptic variances $g_{pq}$ differ across populations, the covariances $\sigma_{nm}$ are different for excitatory and inhibitory populations even if the reciprocal correlations are uniform (Fig 3L and 3M).

These results directly extend to networks with heterogeneous reciprocal correlations $\eta_{pq}$, $p$, $q = E, I$ (Fig 4). Finite-size simulations in this circumstance show the existence of additional, complex conjugate outliers accurately predicted by the cubic term in Eq (8) (Fig 4G, coloured circles and triangles are overlapping, containing outlier scatters at conjugate positions), and this is particularly true in the case where the impact of higher-order structures is marginal compared to that of structures up to the second-order. Moreover, the covariances $\sigma_{nm}^p$ between the entries of low-rank connectivity vectors in this case differ between the excitatory and inhibitory population.

## 1.5 Approximating low-dimensional dynamics for locally-defined connectivity

In previous sections, we developed a rank-one approximation of locally-defined excitatory-inhibitory connectivity. Here we use this approximation to describe the resulting low-dimensional dynamics. We consider networks of rate units, where the activation $x_i$ of unit $i$ obeys

$$\dot{x}_i(t) = -x_i(t) + \sum_{j=1}^{N} J_{ij}\phi(x_j(t)). \tag{14}$$

Here $\phi(x) = 1 + \tanh(x - \theta)$ is a positive transfer function, and for simplicity, we focus on autonomous dynamics without external inputs. We start from a locally-defined excitatory-inhibitory connectivity matrix, and compare the resulting activity with the theoretical predictions of our rank-one approximation, for which the dynamics are low-dimensional and analytically tractable. We first summarize the theoretical predictions for those dynamics, and then examine the specific cases of independent and reciprocally-correlated connectivity.

Recent works have showed that in networks with a rank $R$ connectivity matrix, the trajectories $\mathbf{x}(t) = \{x_i(t)\}_{i=1...N}$ are confined to a low-dimensional subspace of the $N-$dimensional space describing the activity of all units [30–33]. In absence of external inputs, this subspace is $R$-dimensional and spanned by the set of connectivity eigenvectors $\mathbf{m}^{(r)}$ for $r = 1...R$, so that the trajectories can be parametrized as $\mathbf{x} = \sum_{r=1}^{R} \kappa_r \mathbf{m}^{(r)}$ where $\kappa_r$ is a collective latent variable representing activity along $\mathbf{m}^{(r)}$. For a rank-one ($R = 1$) connectivity corresponding to an

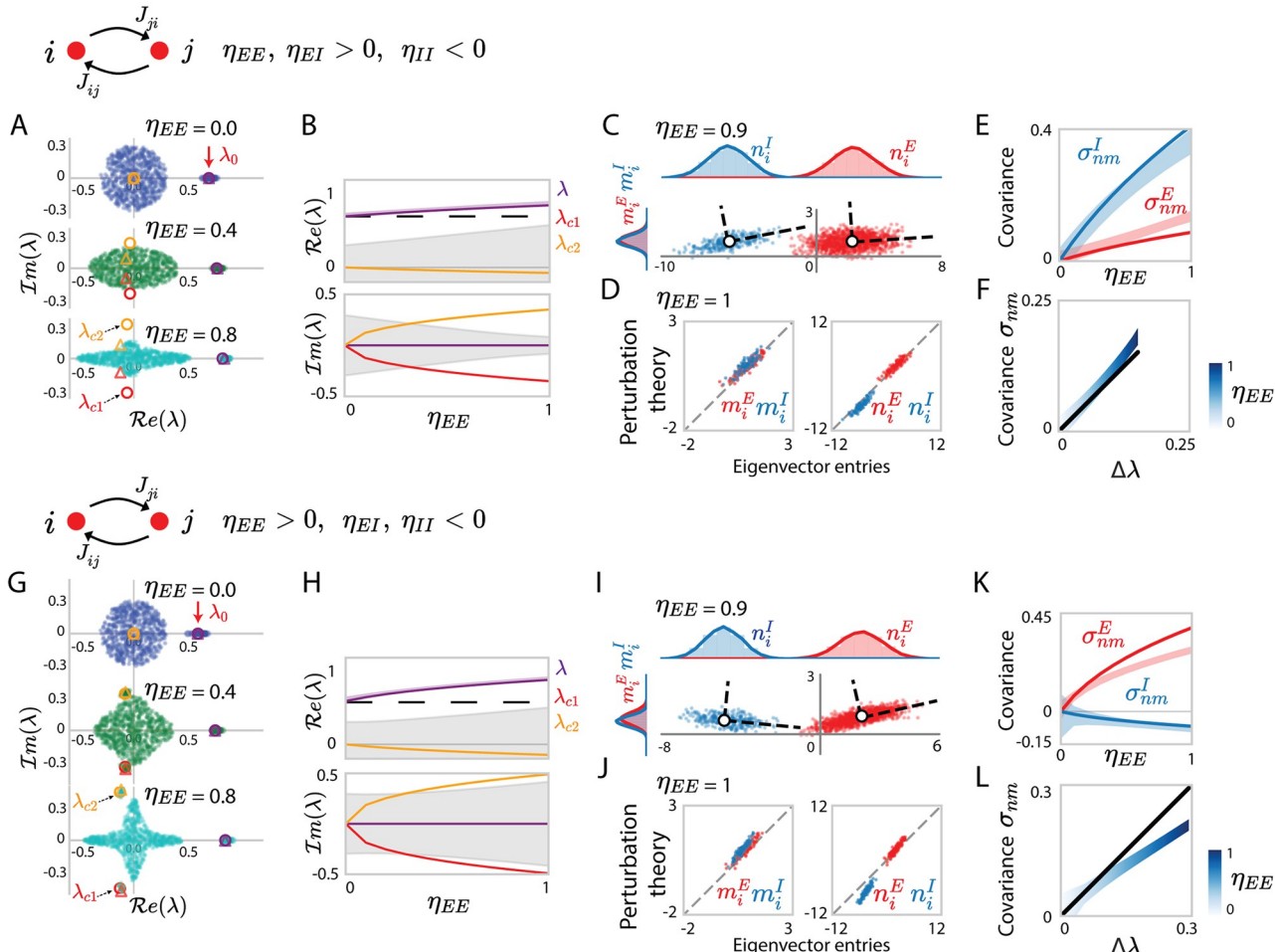

**Fig 4. Eigenvalues and dominant eigenvectors for network connectivity matrix with *heterogeneous* reciprocal correlations.** (A) Eigenvalue spectra of excitatory-inhibitory connectivity matrices **J**, with homogeneous variance parameter $g = 0.3$ but cell-type-dependent reciprocal correlations $\eta_{EE} = \eta_{EI} = -\eta_{II} > 0$, and $\eta_{EE}$ increasing from 0 to 1. Different colours from top to bottom correspond to networks with different values of $\eta_{EE}$. The dots in the elliptical bulk show 600 eigenvalues for one realization of the random connectivity. Outlying eigenvalues are shown for 30 realizations of the random connectivity, the dispersion reflects finite-size effects. The red arrow on the top points to the eigenvalue $\lambda_0$ of the mean connectivity $\bar{\mathbf{J}}$. Coloured circles are the eigenvalues predicted using determinant lemma and truncated series expansion (see Eqs (7) and (8), truncating at $k = 2$), coloured triangles are the eigenvalues predicted using determinant lemma without finite truncation (see S6 Text, Eqs. (153)-(157)). (B) Comparison of the eigenvalues from the finite-size simulation with the predictions of the determinant lemma as the reciprocal correlation $\eta_{EE}$ ($-\eta_{II}$) is increased. The coloured solid lines show the roots of the third-order polynomial in Eq (8) (truncated series expansion). The light purple area indicates the empirical distribution of the dominant outlier for 30 realizations, reflecting finite-size effects; while the black dashed line is the unperturbed eigenvalue $\lambda_0$. The grey areas represent the areas covered by the eigenvalue bulk. (C) Scatter plot showing for each neuron $i$ its entry $n_i$ on the left eigenvector against its entry $m_i$ on the right eigenvector. Red and blue colours represent respectively excitatory and inhibitory neurons. The white dots and the dashed lines respectively indicate the means and covariances for each population. (D) Comparison between eigenvector entries obtained from direct eigen-decomposition of **J** with projections obtained using perturbation theory (Eqs (9) and (10)) in a given random connectivity **Z**. (E) Comparison between simulations (coloured areas, finite-size effects) and theoretical predictions (coloured lines, Eq (97)) for the population covariance $\sigma_{nm}^p$ of the entries on the left and right connectivity eigenvectors to different populations. (F) Comparison of the overall covariance $\sigma_{nm}$ (Eq (72)) with the deviation $\Delta\lambda$ of the dominant outlying eigenvalue from the unperturbed value $\lambda_0$. Empirical covariance (gradient blue area reflects finite-size effects, where the colour depth stands for $\eta$) compared with the theoretical prediction (black line) obtained using Eqs (97) and (92). The x-axis uses the theoretical prediction of the deviation of the eigenvalue $\lambda$ from $\lambda_0$. (G-L) Same as (A-F) for a connectivity matrix with heterogeneous reciprocal correlations: $\eta_{EE} = -\eta_{EI} = -\eta_{II} > 0$, and $\eta_{EE}$ increasing from 0 to 1. In (B, E, F, H, K. L), the departure of the centres of the numerical simulation results from the theoretical predictions reflect the systematic errors due to the first-order perturbation approximation of eigenvalues and eigenvectors. Other network parameters: $N_E = 4N_I = 1200$ and homogeneous variance parameters $g_{pq} = g = 0.3$ in all simulations, $J_E = 2.0$, $J_I = 1.3$ for networks in (A-F), $J_E = 2.0$, $J_I = 1.4$ for networks in (G-L).

approximation of our locally-defined E-I network, the dynamics can therefore be represented by a single latent variable $\kappa$, so that the activation of unit $i$ is given by

$$x_i(t) \quad = \quad \kappa m_i \tag{15}$$

$$= \quad \kappa \bar{m}_i + \kappa \Delta m_i, \tag{16}$$

where we inserted the expression for $m_i$ obtained from a first-order perturbation (Eqs (9) and (10)). Note that since $\bar{m}_i = 1$, the first term in the r. h. s. of Eq (16) corresponds to population-averaged mean activity $\mu_x^p$, while the second term is the deviation of the activity of unit $i$ from the population average, which statistically leads to the population-averaged variance $\Delta_x^p$ of neuronal activations (Methods Sec. 2.6.1, Eq (111))

$$\begin{aligned} \mu_x^p \quad &= \kappa \\ \Delta_x^p \quad &= \kappa^2 \sigma_{m^p}^2. \end{aligned} \tag{17}$$

By inserting the values for $\Delta m_i$ obtained using specific realizations of random connectivity in Eq (10) into Eq (16), the rank-one approximation accounts for the heterogeneous firing activity of single units coming from the synaptic weights in specific instances of locally-defined networks **Z**. Moreover, the rank-one theory predicts that both the population-averaged mean and the standard deviation of activations in the network are proportional to $\kappa$.

The values taken by the latent variable $\kappa$ can be determined by projecting Eq (14) onto **m** and inserting Eq (16) (Methods Secs. 2.6.1, 2.6.2). This leads to a closed equation for the dynamics of $\kappa(t)$:

$$\dot{\kappa} = -\kappa + \sum_{p=E,I} \alpha_p \left( \bar{n}^p \langle \phi(\kappa \bar{m}^p, \kappa^2 \sigma_{m^p}^2) \rangle + \langle \phi'(\kappa \bar{m}^p, \kappa^2 \sigma_{m^p}^2) \rangle \sigma_{nm}^p \kappa \right), \tag{18}$$

where the brackets denote a Gaussian average (see Eq (112)), $\alpha_p$ is the fraction of neurons belonging to population $p$, $\sigma_{nm}^p$ is the population covariance between elements $m_i$ and $n_i$ with $i$ belonging to population $p$ (Table 1). The steady state then obeys

$$\kappa = F_{mean}(\kappa) + F_{cov}(\kappa) \tag{19}$$

where

$$\begin{aligned} F_{mean}(\kappa) \quad &= \sum_{p=E,I} \alpha_p \bar{n}^p \langle \phi(\kappa \bar{m}^p, \kappa^2 \sigma_{m^p}^2) \rangle, \\ F_{cov}(\kappa) \quad &= \sum_{p=E,I} \alpha_p \langle \phi'(\kappa \bar{m}^p, \kappa^2 \sigma_{m^p}^2) \rangle \sigma_{nm}^p \kappa. \end{aligned} \tag{20}$$

The two terms in the r. h. s. of Eq (19) show that the contributions of recurrent synaptic inputs to the latent dynamics $\kappa$ come from two sources: (i) the population means of the left and right connectivity eigenvectors $\bar{n}^p$ and $\bar{m}^p$ that contribute to $F_{mean}(\kappa)$ (Eqs (54) and (56)); (ii) the covariance $\sigma_{nm}^p$ between the left and right connectivity eigenvectors that contributes only to $F_{cov}(\kappa)$. In the low-rank approximation (Eqs (2) and (41)) of the locally-defined E-I connectivity (Eq (5)), these two terms have distinct origins: the mean comes from the independent components of the connectivity (Eqs (55) and (56)); while the covariance comes from reciprocal correlations between connections (Eqs (96) and (97)). We next examine separately the effects on dynamics of these two connectivity components.

**1.5.1 Independently generated local connectivity.** When synaptic connections are generated independently from a Gaussian distributions based on the identities of pre- and post-

synaptic populations, the rank-one approximation of connectivity leads to uncorrelated left and right connectivity vectors $\mathbf{n}$ and $\mathbf{m}$, so that $\sigma^p_{nm} = 0$ for $p = E, I$. In consequence, only the first term is present in the r. h. s. of Eq (19), and the fixed point of the latent dynamics is given by a difference between excitatory and an inhibitory feedback (Eq (116)):

$$\kappa = J_E \langle \phi(\kappa, \kappa^2 \sigma^2_{m^E}) \rangle - J_I \langle \phi(\kappa, \kappa^2 \sigma^2_{m^I}) \rangle. \tag{21}$$

As long as the mean inhibition $J_I$ is strong enough to balance the mean excitation $J_E$, Eq (21) predicts a single fixed point. As $J_E$ is increased, positive feedback begins to dominate and leads to a bifurcation to a bistable regime for the latent dynamic variable $\kappa$ (Fig 5A–5C, S2 Text).

This bistability due to positive feedback is expected on the basis of mean connectivity alone. Indeed, replacing the connectivity matrix by its mean $\bar{\mathbf{J}}$ is equivalent to a rank-one approximation with $\mathbf{m} = \bar{\mathbf{m}}$ and $\mathbf{n} = \bar{\mathbf{n}}$ which lead to Eq (115) with $\sigma^2_{m^p} = 0$ for $p = E, I$, this reduced model actually corresponds to the classic Amari-Grossberg rate model [49–52]. The additional first-order perturbation term in the rank-one approximation (Eqs (9) and (10)) additionally takes into account fluctuations in the connectivity, which leads to a non-zero $\sigma^2_{m^p}$, and modifies the fixed points predicted by Eq (21). In consequence, the bifurcation to bistability takes place at higher values of $J_E$ than predicted from mean connectivity alone (purple lines compared to orange lines in Fig 5C).

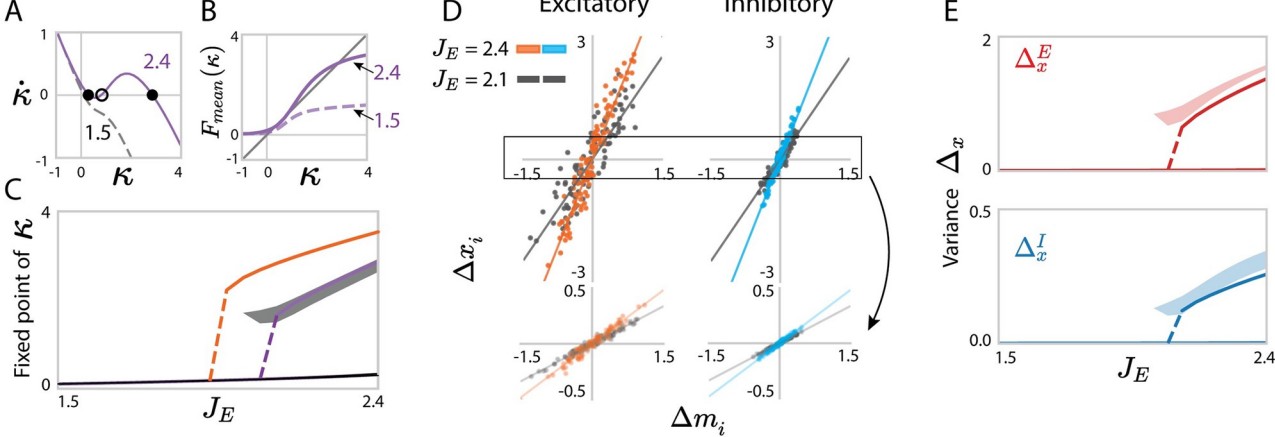

**Fig 5. Predicting low-dimensional dynamics using a rank-one approximation of networks with independent Gaussian connectivity.** (A) Fixed points of the latent variable $\kappa$ in the rank-one approximation. The lines show the dynamics $\dot{\kappa}$ as function of $\kappa$, predicted by Eq (18) (solid line: $J_E = 2.4$; dashed line: $J_E = 1.5$). The intersections with $y = 0$ correspond to fixed points (filled dots: stable; unfilled dot: unstable). (B) Contribution of mean connectivity to the latent dynamics, $F_{mean}(\kappa)$ in Eq (20), for two values of $J_E$. (C) Bifurcation diagram for increasing $J_E$: analytical predictions of Eq (21) compared with simulations of the full network with locally-defined connectivity. orange line: analytical prediction including only the mean part of the connectivity ($\sigma^2_{m^p} = 0$ in Eq (21)); purple line: analytical prediction including the first-order perturbation term in the rank-one approximation; gray: projection of simulated activity $\mathbf{x}$ onto the connectivity vector $\mathbf{m}$ computed by perturbation theory for 30 realizations of random connectivity $\mathbf{Z}$, the shaded area reflects finite-size effects (Eq (104)). (D) Comparison between simulations and mappings obtained from the low-rank approximation (Eqs (10) and (16)) for the activity of individual units in a given realization of the random connectivity $\mathbf{Z}$. For each unit $i$, a dot shows the deviation $\Delta x_i$ of its steady-state activity from the population average, against its value $\Delta m_i$ of the perturbed part of the connectivity vector $\mathbf{m}$ (Eq (16)). The low-rank theory predicts $\Delta x_i = \kappa \Delta m_i$. Orange, cyan and gray scatters show excitatory or inhibitory populations, each for several values of $J_E$. Lines represent $y = \kappa x$, where $\kappa$ is obtained from Eq (21). Upper panels show the result in a realization with a high fixed point, bottom panels show the result in a realization with a low fixed point. (E) Comparison between the predictions (solid lines) and simulations (shaded areas) for the population-averaged variances of $\Delta x_i$. Shaded areas show mean±std and reflect finite-size effects. In C, E, the departure of the centres of the numerical simulation results from the theoretical predictions reflect the systematic errors due to the approximation of the latent dynamical variable $\kappa$ as well as the low-rank connectivity statistics. Other network parameters: $N_E = 4N_I = 1200$, $J_I = 0.6$, $g_{EE}$: $g_{EI}$: $g_{IE}$: $g_{II} = 1.0$: $0.5$: $0.2$: $0.8$ and $g_{EE} = 0.8$. The transfer function $\phi$ has parameter $\theta = 1.5$.

More importantly, we find that the first-order perturbation in the rank-one approximation accurately accounts for the heterogeneous firing rates of individual neurons for specific instances of the random, locally-defined connectivity $\mathbf{Z}$ (Eqs (16) and (69), Fig 5D), and therefore predicts well the variance across neurons of the steady state of population dynamics $\Delta_x^p$. In particular, cell-type dependent variances in the synaptic connectivity, lead to distinct variances $\Delta_x^E$ and $\Delta_x^I$ for excitatory and inhibitory populations (Eqs (90) and (130), Fig 5E).

Note that the independently generated local connectivity can be treated analytically without resorting to a rank-one approximation, by using a different variant of mean-field theory originally developed for randomly connected networks. [30, 53]. That theory is not perturbative, and takes into account an additional term in the variance (see Methods Sec. 2.6.3 and S3 Text for more details). However, in contrast to the rank-one approximation, it does not predict the activity of individual neurons, and is challenging to extend beyond independent random connectivity.

**1.5.2 Reciprocal motifs.** We next turn to the predictions of the rank-one approximation for dynamics resulting from locally-defined connectivity with reciprocal motifs. In this case, the additional reciprocal correlations in the random part of the connectivity lead to a non-zero covariance $\sigma_{nm}$ between the connectivity vectors $\mathbf{n}$ and $\mathbf{m}$ in the rank-one approximation (Eqs (92) and (97)). This covariance in turn generates an additional feedback component in the dynamics of the latent variable, the second term in the r. h. s. of Eq (19).

Specifically, the excitation-dominated dynamical regime is largely determined by the noiseless mean connectivity $\bar{\mathbf{J}}$, but beyond this mean approximation, positive reciprocal correlations combined with the excitation-dominated connectivity enhance positive feedback with respect to mean connectivity alone (Methods Sec. 2.5.2, Eq (97)). As a result, progressively increasing the reciprocal correlations can therefore induce a bifurcation to bistability, even if the mean excitation provided by the mean approximation $\bar{\mathbf{J}}$ is not sufficient by itself to support two stable states (Fig 6A–6C). This is a major novel effect of reciprocal motifs on collective dynamics. As in the case of independent connectivity, we moreover find that the perturbative term in the rank-one approximation accounts for the heterogeneous activity of individual neurons in specific realizations of the connectivity $\mathbf{Z}$ (Fig 6D, Eqs (16) and (69)), and therefore also predicts the cell-type dependent variances of activity (Fig 6E).

More generally, our rank-one approximation allows us to describe the latent dynamics when the degree of reciprocal correlation depends across the pre- and post-synaptic populations (Results Sec. 1.4). Such heterogeneity in reciprocal correlations can enhance different types of feedback. For example, antisymmetric connectivity within inhibitory populations ($\eta_{II}$ < 0) disinhibits excitatory population and thus facilitates bistable transitions (Fig 7A–7C) compared to networks with homogeneous reciprocal correlations. In contrast, excitation-dominated connectivity with homogeneous negative reciprocity ($\eta$ < 0) generate negative feedback and therefore suppress the global dynamics from bistable state to quiescent (Fig 7D–7F).

Importantly, describing the role of reciprocal correlations on latent dynamics relies on our global low-rank approximation of locally-defined connectivity. In particular, the effects of such correlations cannot be captured by considering only mean connectivity and population-averaged activity (first term in the r. h. s. of Eq (19)). Moreover, including reciprocal correlations in classical mean-field approaches to randomly connected networks is technically challenging [18].

## 1.6 Extension: E-I networks with sparse connectivity

In previous sections, we examined locally-defined connectivity generated using Gaussian distributions of individual synaptic weights (function $f^{pq}$ in Eq (1)). Our results for the low-rank

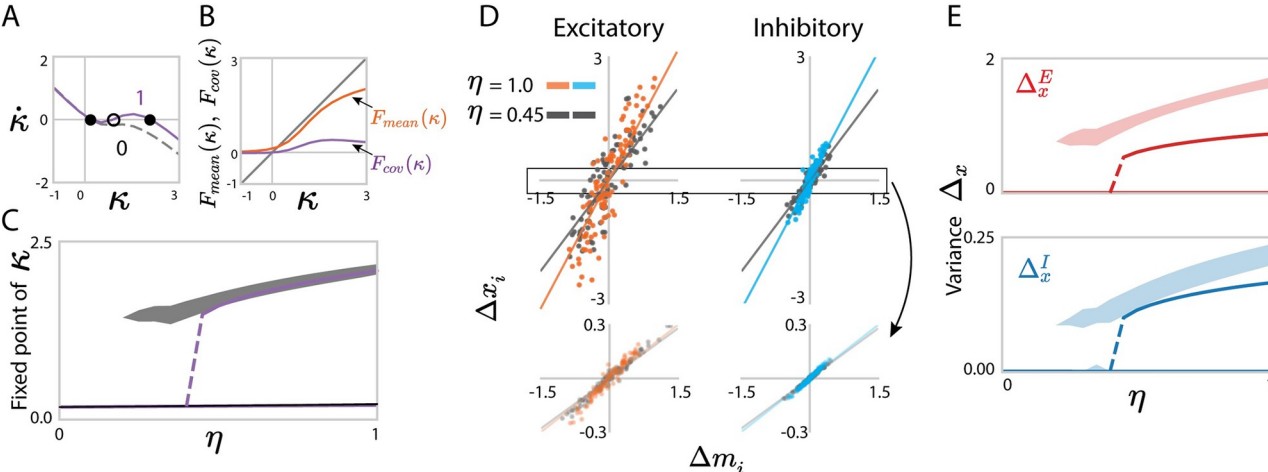

**Fig 6. Predicting low-dimensional dynamics using a rank-one approximation of networks with homogeneous reciprocal motifs.** (A) Influence of reciprocal correlations on fixed points of the latent variable $\kappa$ in the rank-one approximation. The lines show the dynamics $\dot{\kappa}$ as function of $\kappa$, predicted by Eq (18) (solid line: $\eta = 1$; dashed line: $\eta = 0$). The intersections with $y = 0$ correspond to fixed points (filled dots: stable; unfilled dot: unstable). (B) Comparison of the contributions of mean connectivity $F_{mean}(\kappa)$ and covariance $F_{cov}(\kappa)$ to the latent dynamics of $\kappa$ (Eq (20)) for $\eta = 1$. (C) Bifurcation diagram for increasing $\eta$ at fixed $J_E$. Solid purple lines: analytical predictions of Eqs (19) and (20); gray areas: projection of simulated activity **x** onto the connectivity vector **m** computed by perturbation theory (Eq (104)) for 30 realizations of random connectivity **Z**, the shaded area reflects finite-size effects. (D) Comparison between simulations and mappings obtained from the low-rank approximation for the activity of individual units in a given realization of the random connectivity **Z**. For each unit $i$, a dot shows the deviation $\Delta x_i$ of its steady-state activity from the population average, against its value $\Delta m_i$ of the perturbed part of the connectivity vector **m** (Eq (16)). The low-rank theory maps $\Delta x_i = \kappa \Delta m_i$. Orange, cyan and gray scatters show excitatory or inhibitory populations, each for two values of $\eta$. Lines represent $y = \kappa x$, where $\kappa$ is obtained from Eqs (19) and (20). Upper panels show the result in a realization with a high fixed point, bottom panels show the result in a realization with a low fixed point. (E) Comparison between the predictions (solid lines, Eq. (134)) and simulations (shaded areas) for the population-averaged variances of $\Delta x_i$. Shaded areas show mean±std and reflect finite-size effects. In C, E, the departure of the centres of the numerical simulation results from the theoretical predictions reflect the systematic errors due to the approximation of the latent dynamical variable $\kappa$ as well as the low-rank connectivity statistics. Other network parameters: $N_E = 4N_I = 1200$, $J_E = 1.9$, $J_I = 0.6$, $g_{EE}$: $g_{EI}$: $g_{IE}$: $g_{II} = 1.0$: 0.5: 0.2: 0.8 and $g_{EE} = 0.8$. The transfer function $\phi$ has parameter $\theta = 1.5$.

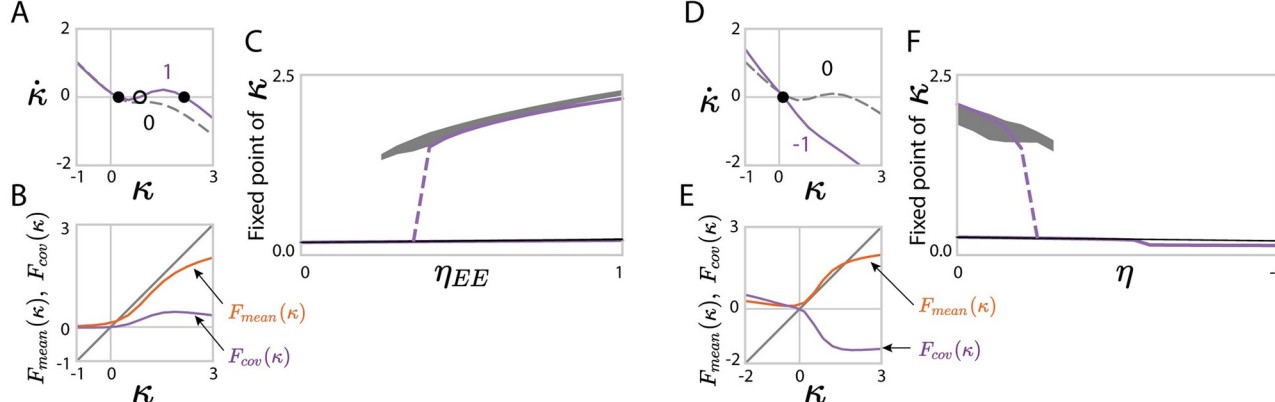

**Fig 7. Predictions for low-dimensional dynamics using a rank-one approximation of networks with non-homogeneous and anti-symmetric reciprocal motifs.** (A-D) Fixed points of latent dynamics in networks with heterogeneous, cell-type-dependent reciprocal correlations: $\eta_{EE} = \eta_{EI} = -\eta_{II}$. Network parameters: $N_E = 4N_I = 1200$, $g_{EE}$: $g_{EI}$: $g_{IE}$: $g_{II} = 1.0$: 0.5: 0.2: 0.8, $g_{EE} = 0.8$, $J_E = 1.9$ and $J_I = 0.6$. (E-H) Fixed points of latent dynamics in networks with homogeneously anti-symmetric motifs, $\eta_{pq} = \eta \in [-1, 0]$. Network parameters: $N_E = 4N_I = 1200$, $g_{EE}$: $g_{EI}$: $g_{IE}$: $g_{II} = 1.0$: 0.5: 0.2: 0.8, $g_{EE} = 0.8$, $J_E = 2.2$ and $J_I = 0.6$. In C and F, gray areas show projections of simulated activity **x** onto the connectivity vector **m** computed by perturbation theory (Eq (104)), shaded areas show mean±std and reflect finite-size effects. In C, F the departure of the centres of the numerical simulation results from the theoretical predictions reflect the systematic errors due to the approximation of the latent dynamical variable $\kappa$ as well as the low-rank connectivity statistics.

approximation of locally-defined connectivity are however independent of the precise form of the distribution $f^{pq}$. In particular, our finding that the resulting low-rank structure obeys Gaussian-mixture statistics is universal, in the sense that it is valid for any distribution $f^{pq}$ for which the central limit theorem holds (see Discussion). To illustrate this universality, here we turn to networks with sparse connectivity, generated from Bernoulli distributions $f^{pq}$ taking values 0 and $A_q$ (where $A_q = A_E, A_I$ refer to strengths of excitatory and inhibitory connections), with a uniform fraction $c$ of non-zero connections (see Eq (28)). In this case, the variance of the synaptic strengths $J_{ij}$ scales as $1/N$, as we assumed for Gaussian connectivity. The statistics of the corresponding rank-one approximation are fully determined by the mean, variance and covariance of synaptic weights in the excitatory and inhibitory populations. Here we compare the predictions of our perturbative approximation with direct simulations of full-rank networks with locally-defined connectivity. We first consider independently generated connectivity, and then turn to reciprocal motifs.

**1.6.1 Independently generated sparse connectivity.** For networks with independently generated sparse connectivity, the mean and variance of individual synaptic weights are given by $cA_q$ and $c(1-c)A_q^2$, where $q = E, I$ refers to the population of the presynaptic neuron. Previous works have shown that for such sparse networks where the variances of synaptic weights scale as $1/N$, the bulk of the eigenvalues are distributed within a circle in the complex plane with a radius $r_g$ determined by Eq. (149) (Fig 8A) [54, 55], as expected from the universality theorem for random matrices [56]. The overall mean of the synaptic weights instead determines the mean outlying eigenvalue (Eq (78), Fig 8A). In sparse networks, the main novelty with respect to the Gaussian case is that the mean and variance of synaptic weights are not independent parameters, but are instead both set by the synaptic strengths $A_E$ and $A_I$ as well as the network's sparsity $c$ (Eqs (38) and (39)). In consequence, varying these couplings changes both the radius of the bulk and the outlying eigenvalue, and can lead to intersections where the outliers dip into the bulk (see S4 Text for details).

As expected, as long as the outlier lies outside of the eigenvalue bulk, the statistics of entries on the resulting low-rank approximating eigenvectors are well described by a Gaussian-mixture distribution with parameters fully determined by the mean and variance of the synaptic weights (Eq (91), Fig 8B). Our perturbative approximation Eq (69) as well, accounts for the individual entries of the right- and left-eigenvectors corresponding to the outlier from the specific realization of the fluctuation matrix **Z** (Fig 8C).

Our predictions for the low-dimensional dynamics based on the rank-one approximation therefore directly extend to sparse networks. Comparing with direct simulations, we found that Eq (117) predicts well the global latent variable $\kappa$ obtained by projecting the activity **x** onto the approximated rank-one eigenvector **m** (Eq (104)). As the E/I ratio is increased, positive feedback increases, and the latent variable $\kappa$ undergoes a transition from a single fixed point to two bistable states (Fig 8D).

From the statistics of the right eigenvector **m** (Eq (91)), our analysis predicts the heterogeneity of activity in terms of population-averaged variances $\Delta_x^{E/I}$ (Fig 8E). This heterogeneity is identical in excitatory and inhibitory populations, as their right eigenvectors $\mathbf{m}^E$, $\mathbf{m}^I$ have identical fluctuations (Fig 8B and 8C). For individual realizations of the sparse connectivity, the rank-one approximation $\mathbf{x} = \kappa\mathbf{m}$ moreover captures the activation $x_i$ of individual neurons in the specific simulation instances (Fig 8F).

**1.6.2 Sparse EI networks with reciprocal motifs.** For sparse E-I networks, we generate reciprocal motifs by introducing a fraction $\rho_{pq}$ of reciprocally connected pairs of neurons. Together with the sparsity $c$ and the synaptic strengths, the parameter $\rho_{pq}$ determines the cell-type dependent reciprocal correlation $\eta_{pq}$ (Methods Sec. 2.1.2 Eqs (34) and (39), Fig 9A).

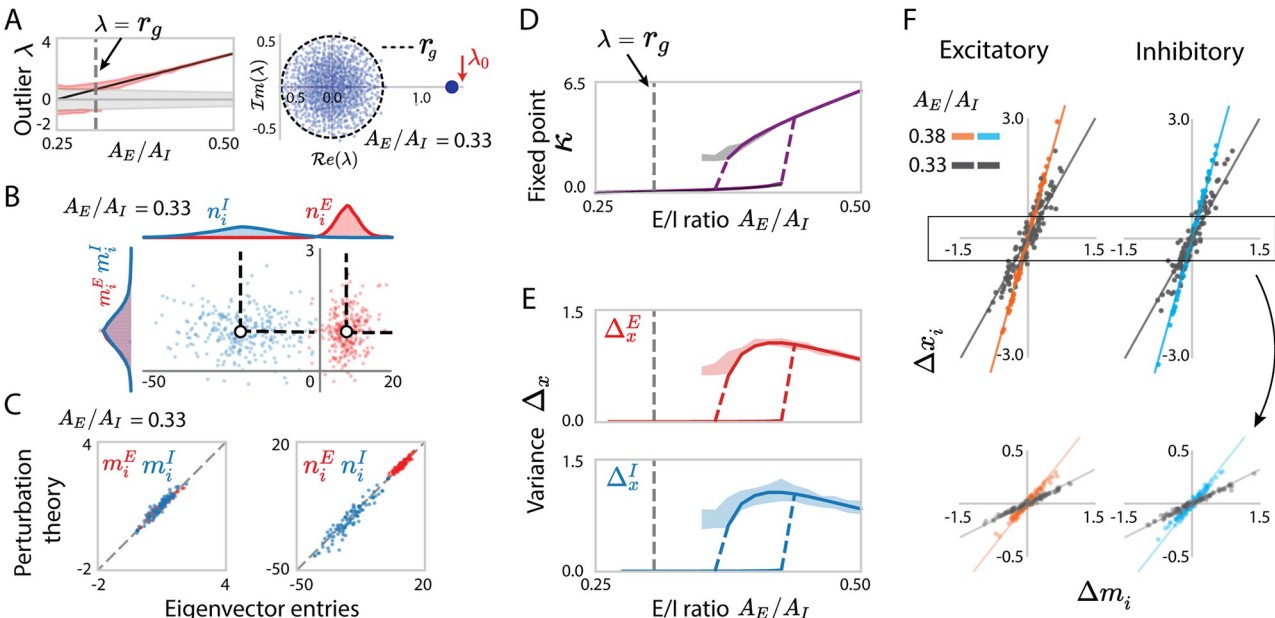

**Fig 8. Rank-one approximation and predicted low-dimensional dynamics for sparse excitatory-inhibitory networks.** (A) Left: Comparison of the predicted eigenvalue outlier $\lambda_0 = c(N_E A_E + N_I A_I)$ (black line) with finite-size simulations (red area shows mean±std for 30 realizations and reflects finite-size effects). The gray area represents the area covered by the eigenvalue bulk. Right: example spectrum of one realization of connectivity matrix with E/I ratio $A_E/A_I = 0.33$, and the radius $r_g$ of the eigenvalue bulk computed from the statistically equivalent Gaussian connectivity (see S4 Text). (B) Scatter plot showing for each neuron $i$ its entry $n_i$ on the left eigenvector against its entry $m_i$ on the right eigenvector. Red and blue colours represent respectively excitatory and inhibitory neurons. The white dots and the dash lines respectively indicate the means and covariances for each population obtained from simulations. For visualization purposes, the $x-$ and $y$-axis are scaled unequally. (C) Comparison between eigenvector entries obtained from direct eigen-decomposition of **J** with projections obtained using perturbation theory (Eqs (9) and (10)) in a given realization of the sparse connectivity **J**. (D) Bifurcation diagram for increasing the ratio $A_E/A_I$: analytical predictions of Eq (117) compared with simulations of the full network with locally generated sparse connectivity. Purple line: analytical prediction including the first-order perturbation term in the rank-one approximation; gray: projection of simulated activity onto the connectivity vector **m** computed by perturbation theory Eq (104) for 30 realizations of sparse connectivty **J**, the shaded area reflects finite-size effects. (E) Comparison between the predictions (solid lines) and simulations (shaded areas) for the population-averaged variances of $\Delta x_i$. Shaded areas show mean±std and reflect finite-size effects. (F) Comparison between simulations and mappings obtained from the low-rank approximation for the activity of individual units in a given realization of the sparse connectivity **J**. For each unit $i$, a dot shows the deviation $\Delta x_i$ of the steady-state activity from the population average, against the corresponding value $\Delta m_i$ of the perturbed part of the connectivity vector **m** (Eq (16)). The low-rank theory maps $\Delta x_i = \kappa \Delta m_i$. Orange, cyan and gray scatters show excitatory or inhibitory populations, each for two values of the ratio $A_E/A_I$. Lines represent $y = \kappa x$, where $\kappa$ is obtained from Eqs (19) and (20). Upper panels show the result in a realization with a high fixed point, bottom panels show the result in a realization with a low fixed point. The gray vertical dashed line in A left, D, E correspond to the critical point $c$ at which the absolute value of the outlier is equal to the radius of the eigenvalue bulk. In A, D, E the departure of the centres of the numerical simulation results from the theoretical predictions reflect the systematic errors due to the approximation of the low-rank connectivity statistics as well as the latent dynamical variable $\kappa$. Network parameters: $N_E = 4N_I = 800$, $c = 0.3$, $A_E = 0.025$. The transfer function $\phi$ has parameter $\theta = 1.5$.

We first examine the effect of the reciprocal motifs on the statistical properties of the eigenvalues and eigenvectors. The spectrum still consists of continuous eigenvalues and a discrete outlier. Since in the independent case the outlier depends only on $A_{E/I}$ and sparsity $c$, here we fix these two variables but increase $\eta_{EE} = -\eta_{EI}$ while keeping $\eta_{II}$ constant. As in the case of dense, Gaussian networks, the outlier increases with the increasing reciprocal correlation and deviates from the outlier of the corresponding independent sparse connectivity matrix (Fig 9B and 9C). Moreover, in the large network limit, we find that if means, variances, and reciprocal correlations are identical, dense Gaussian connectivity leads to the same eigenvalue spectrum as the sparse connectivity (Methods Secs. 2.1.2, 2.1.3, Eqs (38) and (39)). We furthermore mathematically predict two additional conjugate eigenvalue outliers generated by the reciprocal connections in the sparse case (Eqs (82), (88) and (89), Fig 9B).

As for uncorrelated connectivity, perturbation theory accounts for the individual left and right eigenvector entries for specific instances of the sparse connectivity, which altogether

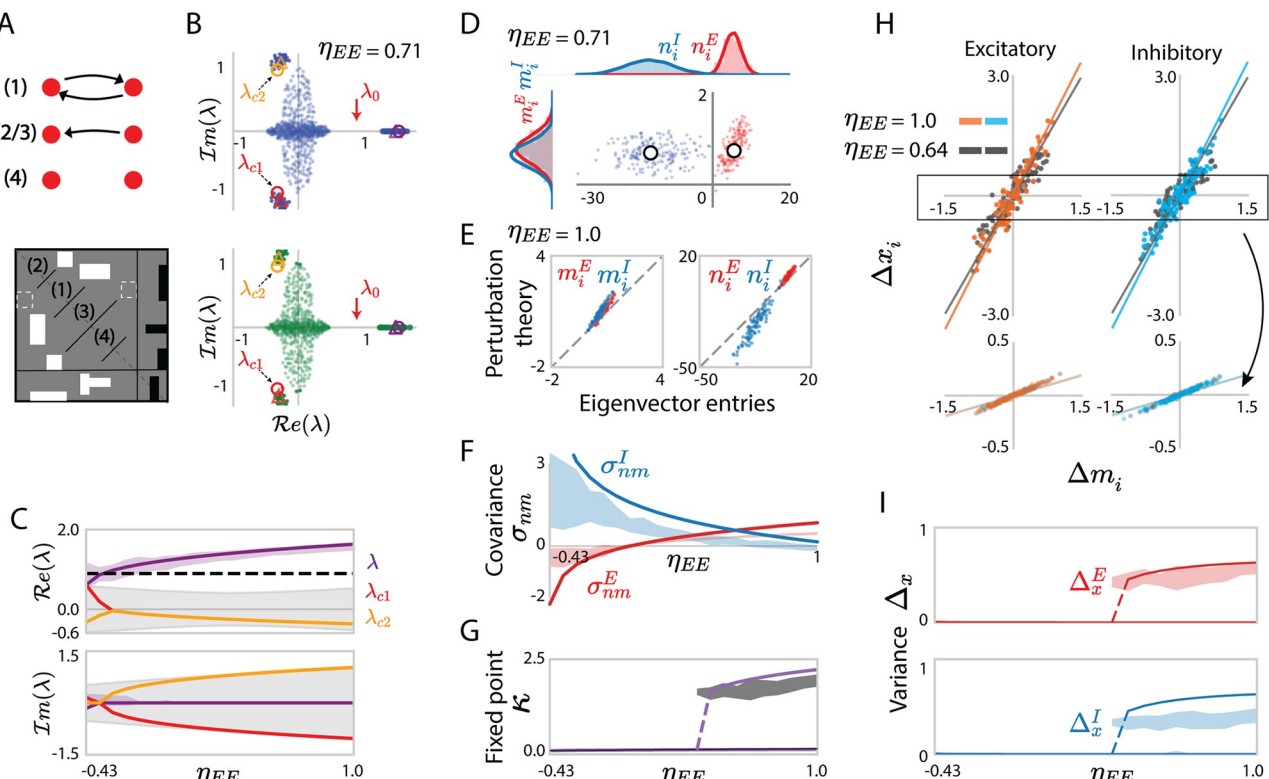

**Fig 9. Characterizing connectivity statistical properties and low-dimensional dynamics for the sparse network with reciprocal motifs.** (A) Schematics of a sparse EI network with four forms of paired connections. White and black rectangles represent the non-zero excitatory and inhibitory sparse connections. (B) Eigenvalue spectrum of the sparse connectivity (upper panel) and from the equivalent Gaussian connectivity (bottom panel) with reciprocal motifs. Cell-type-dependent reciprocal correlations are $\eta_{EE} = 0.71$, $\eta_{EI} = -0.71$, $\eta_{II} = -0.43$ in both connectivity matrices, continuous eigenvalue bulks show eigenvalues for one realization of the network connectivity. Red arrows point to the unperturbed eigenvalue $\lambda_0$. Outlying eigenvalues are shown for 30 realizations of the network connectivity. Coloured circles are the eigenvalues predicted using determinant lemma and truncated series expansion (see Eqs (7) and (8), truncating at $k = 2$), coloured triangles are the eigenvalues predicted using determinant lemma without finite truncation (see S6 Text, Eqs. (153)-(157)). (C) Comparison of the eigenvalues from the finite-size simulation of the sparse connectivity, with the predictions of the determinant lemma as progressively increasing the reciprocal correlation $\eta_{EE}$ ($-\eta_{EI}$). The coloured solid lines show the roots of the third-order polynomial in Eq (8). The purple area indicates the empirical distribution (mean±std, finite-size effects) of the dominant outlier for 30 realizations of sparse connectivity $\mathbf{J}$, while the black dashed line is the eigenvalue $\lambda_0$ of the corresponding independent sparse connectivity matrix (Eq (78)). The gray areas correspond to the areas covered by the eigenvalue bulk. (D) Scatter plot showing for each neuron $i$ its entry $n_i$ on the left eigenvector against its entry $m_i$ on the right eigenvector. Red and blue colours represent respectively excitatory and inhibitory neurons. The white dots indicate the means for each population obtained from simulations. For visualization purposes, the $x$- and $y$-axis are scaled unequally. (E) Comparison between eigenvector entries obtained from direct eigen-decomposition of $\mathbf{J}$ with projections obtained using perturbation theory (Eqs (9) and (10)) in a given realization of the sparse connectivity $\mathbf{J}$. (F) Comparison between the population covariance $\sigma_{nm}^p$ of the entries on the left and right connectivity eigenvectors to different populations (coloured areas, finite-size effects) and the predictions of perturbation theory (coloured lines, Eq (99)). (G) Bifurcation diagram for increasing the reciprocal correlation $\eta_{EE}$ ($-\eta_{EI}$): analytical predictions of Eq (119) compared with simulations of the full network with locally generated sparse connectivity and reciprocal motifs. Purple line: analytical prediction including the first-order perturbation term in the rank-one approximation; gray: projection of simulated activity onto the connectivity vector $\mathbf{m}$ computed by perturbation theory Eq (104) for 30 realizations of sparse connectivity $\mathbf{J}$, the shaded area reflects finite-size effects. (H) Comparison between simulations and mappings obtained from the low-rank approximation for the activity of individual units in a given realization of the sparse connectivity. For each unit $i$, a dot shows the deviation $\Delta x_i$ of the steady-state activity from the population average, against its value $\Delta m_i$ of the perturbed part of the connectivity vector $\mathbf{m}$ (Eq (16)). The low-rank theory maps $\Delta x_i = \kappa \Delta m_i$. Orange, cyan and gray scatters show excitatory or inhibitory populations, each for two values of the reciprocal correlation $\eta_{EE}$. Lines represent $y = \kappa x$, where $\kappa$ is obtained from Eqs (19) and (20). Upper panels show the result in a realization with a high fixed point, bottom panels show the result in a realization with a low fixed point. (I) Comparison between the predictions (solid lines) and simulations (shaded areas) from the population-averaged variances of $\Delta x_i$, shaded areas show mean±std and reflect finite-size effects. In C, F, G, I, the reciprocal correlations $\eta_{EE} = -\eta_{EI}$ progressively increase from $-0.43$ to $1.0$ while keeping $\eta_{II} = -0.43$ constant ($\rho_{EE} = \rho_{EI}$ increase from 0 to 1 and $\rho_{II} = 0$ is fixed). In C, F, G, I, the departure of the centres of the numerical simulation results from the theoretical predictions reflect the systematic errors due to the approximation of the low-rank connectivity statistics as well as the latent dynamical variable $\kappa$. Network parameters: $N_E = 4N_I = 800$, $c = 0.3$, $A_E = 0.023$, $A_E/A_I = 0.3$. The transfer function $\phi$ has parameter $\theta = 1.5$.

follow Gaussian statistics as expected (Fig 9D and 9E, Eqs (69) and (70)). Importantly, reciprocal correlations induce a non-zero covariance $\sigma_{nm}$ between the entries $m_i$ and $n_i$ of the right and left eigenvectors (Eq (99), Fig 9F).

Finally, we examine the population dynamics in the sparse network with reciprocal motifs using the low-rank approximation derived above. The reciprocal motifs in the example network generate an overall positive feedback. Therefore, gradually increasing the reciprocal correlation $\eta_{EE}$ ($-\eta_{EI}$) in the example network induces a bifurcation into bistability (Eq (119), Fig 9G). Analogous to the projections depicting the individual eigenvector entries obtained using perturbation theory (Methods Sec. 2.3 Eq (69)), the low-rank approximation analytically accounts for the activity of individual neurons in specific connectivity realizations (Eq (16), Fig 9H), and hence the cell-type dependent variances of neuronal activation $\Delta_x^{E/I}$ (Fig 9I) obtained from finite-size simulations of the original sparse networks.

## Discussion

In this work, we unified two different descriptions of connectivity in multi-population networks and thereby connected two broad classes of models. Starting from local statistics of synaptic weights, we approximated the resulting connectivity matrix in terms of a Gaussian-mixture low-rank structure. The obtained, approximate low-rank network model then allowed us to determine the influence of the local connectivity motifs on the global low-dimensional dynamics.

Gaussian-mixture low-rank networks may seem to rely on major simplifying assumptions to make the dynamics more tractable and interpretable, such as Gaussian-distributed entries of the connectivity vectors that are independent across neurons. Our analyses however show that this class of models is nevertheless less restrictive than it may appear at first. Specifically, by using our analytical approach, we show that even if the distribution from which the locally-defined $J_{ij}$ is sampled is not Gaussian (e. g. Bernoulli), the corresponding entries $m_i$, $n_j$ on the connectivity vectors nonetheless converge to a multi-variate Gaussian distribution in the large network limit, provided the conditions of the central limit theorem are met. Importantly, our results for the network connectivity with reciprocal motifs show that the independent assumption for $m_i$ ($n_i$) does not rule out the possibility of reciprocally-correlated synaptic weights ($J_{ij}$, $J_{ji}$) in the locally-defined connectivity.

A key ingredient in our approach is a low-rank approximation of the locally-defined connectivity matrix. Approximating an arbitrary full-rank matrix by a rank-$R$ one is a classical problem in numerical analysis, for which a number of different methods are available depending on the objective of approximation [57]. The most common method is to perform a singular value decomposition (SVD), and keep the top $R$ terms [58]. This method minimizes the Frobenious norm of the difference between the original matrix and its low-rank approximation. Our goal in this study was however to obtain a low-rank approximation that preserves the dominant eigenvalues of the original matrix, as these eigenvalues determine the autonomous dynamics in the network. An SVD-based approximation preserves the top singular values, but in general not the top eigenvalues (S1 Fig), and this can lead to an inaccurate approximation of autonomous dynamics [59]. We therefore opted for an approximation based on truncated eigen-decomposition. When studying input-driven and transient dynamics, different methods for low-rank approximation may be more appropriate, and are a topic of active research [60–65].

To perform the eigen-decomposition of excitatory-inhibitory connectivity matrices, we leveraged the fact that they can be expressed as a sum of a block-like deterministic low-rank matrix and a full-rank random matrix with zero-mean [40]. The eigenspectrum of such

matrices in general consists of a continuously-distributed bulk that is attributed to the random component, and discrete eigenvalues that generated by the low-rank excitatory-and-inhibitory mean connectivity [44, 45]. When the mean of excitatory and inhibitory weights approximately cancel each another, the corresponding eigenvalue is small and resides within the bulk. A number of works have examined the bulk of the eigenvalue spectrum for random matrices [11, 18, 44, 53, 54, 56, 66, 67], and showed that the obtained eigenvalue statistics have important implications for network dynamics such as spontaneous fluctuations [68], oscillations [69, 70] and correlations in asynchronous irregular activity [71]. In contrast, in this work, we focus on the parameter regime where the discrete eigenvalues are outliers and well separated from the eigenvalue bulk. The outlying eigenvalues, and in particular the corresponding eigenvectors have to our knowledge received less attention.

The techniques used in this work on the perturbation eigenvalues can be traced back to the classic work of Tao [45]. The intriguing finding that low-rank perturbations on the i. i. d. matrices have a range of nonlinear effects on the outliers inspired our investigation: If we think of the correlated random connectivity as perturbations on the low-rank dominant structure, what effects do these have on the outliers? Using linear response theory and mean-field techniques, recent research [66] expanded Tao's findings to network dynamics by investigating the outliers of the covariance matrix of the dynamic fluctuations in i. i. d. networks. In particular that work showed the effects of second-order motifs on the eigenvalue bulk distribution of the covariance matrix. Our analyses use the low-rank excitatory-and-inhibitory mean structure to go one step further and analytically demonstrate the impact of cell-type-dependent reciprocal motifs on the outlier λ (Methods Sec. 2.4.2).

The main technical novelty in this work is the use of matrix perturbation theory [47, 72] to approximate the eigenvectors corresponding to the outliers in the eigenspectrum of the locally-defined connectivity matrices. A key output of this approach is the finding that entries of the left- and right-eigenvectors follow multivariate Gaussian distributions, the statistics of which depend on the population the neurons belong to. In particular, these entries on the left and right vectors are uncorrelated in networks with i. i. d. local connectivity (Sec. 1.3); nonetheless, the reciprocal motifs further induce correlations between these entries on the left and right vectors and result in zero-mean overlaps $\sigma_{nm}^p$ between the vectors for each population. (Sec. 1.4). This result provides a general theoretical mapping from locally-defined multi-population models to Gaussian-mixture low-rank networks [31, 33]. It however holds only as long as the entries on the resulting approximation low-rank vectors satisfy the assumption of independent entries across neurons, and the distribution of synaptic weights satisfies the assumptions of the central limit theorem. This specifically rules out, for instance, strong synaptic weights that may be analogous to connectivity hubs, as well as the heavy-tailed distributions often found in experimental studies [1, 73]. Other techniques such as Feynman diagrams [74] may provide a way to further study the effects of structure on dynamics and in particular correlations.

In the networks we considered, the non-random structure in connectivity comes only from the multi-population organization. More specifically, the low-rank skeleton of the locally-defined connectivity matrix is fully specified by the mean synaptic weights between different populations (Eq (3)). This mean connectivity structure largely controls the outlying eigenvalue, and the average values of the corresponding eigenvector entries. The random part of the connectivity and reciprocal motifs can modify the outlying eigenvalue, and add heterogeneity as well as correlations to this underlying structure. The effect of changes to the underlying connectivity is to further regulate the internal network dynamics, i. e., the bistability transition [75]. Particularly, in networks with i. i. d. random connectivity, even though the latent dynamics is primarily determined by the mean excitatory-and-inhibitory connectivity, the

independent random part further determines the heterogeneity of the neural activities and indirectly affects the bifurcation point (Sec. 1.5.1, Fig 5C); however, the reciprocal motifs in the random component directly control the latent dynamics by incorporating recurrent feedback with regard to mean connectivity alone (Sec. 1.5.2). Our perturbative theory moreover allows us to quantify these effects and accurately account for the heterogeneity in dynamics on a single-neuron basis. To further incorporate experimental data on synaptic connectivity into recurrent network models, an important question is how networks of rate units used here relate to more biologically realistic spiking networks where neurons interact through discrete action potentials [76, 77]. One approach for investigating this relationship is to map each spiking neuron onto a rate unit, and therefore compare rate and spiking networks with an identical connectivity matrix. Using this approach, recent work has shown that theoretical results in rate networks directly predict low-dimensional dynamics in spiking networks with identical low-rank connectivity [78]. This provides a possible justification for interpreting the connectivity in rate networks directly in terms of experimentally measured local connectivity statistics.

A key insight from our study is a general relationship between reciprocal motifs in locally-defined connectivity and overlaps among connectivity vectors in low-rank networks, which hasn't been investigated in the previous works of the low-rank model [30, 31, 33, 79]. Indeed, we have shown that correlations between reciprocal synaptic weights generate overlaps beyond the mean in the corresponding low-rank approximation (Eq (95)). Conversely, zero-mean overlaps between connectivity vectors in a low-rank model necessarily imply non-vanishing reciprocal correlations (Eq (43) and S5 Text). Since overlaps between connectivity vectors determine the autonomous recurrent dynamics in low-rank networks, this relationship allowed us to quantify how reciprocal connectivity motifs contribute to network dynamics.

Local statistics of synaptic connectivity are believed to play an important role in the global network dynamics [1, 17, 32, 66]. Our study provides a mathematical theory that relates the local connectivity statistics to global recurrent dynamics through a low-rank approximation. In addition to the reciprocal motifs that we have focused on in this work, it has been demonstrated that other second-order motifs, including convergent, divergent, and chain motifs, have important effects on the statistics of fluctuations [14, 22, 23, 80]. Specifically, recent studies have revealed that the dimensionality of the balanced networks and the norm of the continuous eigenvalue bulk are tuned differently by the statistics of various types of motifs [15, 66, 75]. Examining the effects of these additional types of motifs on eigenvalue outliers and corresponding low-dimensional dynamics would be an important future step to connect models to large-scale electrophysiological recordings of the cortical microcircuits [6, 7, 15, 81].

## 2 Materials and methods

Throughout this study, we consider recurrent networks of $N$ neurons and denote by $\mathbf{J}$ the recurrent connectivity matrix, where $J_{ij}$ is the synaptic strength of the connection from neuron $j$ to neuron $i$.

### 2.1 Locally-defined multi-population connectivity

In this section, we introduce a first class of connectivity models, in which the synaptic couplings are generated based on *local* statistics determined by the identity of pre- and post-synaptic neurons. The $N$ neurons in the network are organized in $P$ populations, where population $p$ has $N_p$ neurons. Denoting by $p$ and $q$ the populations neurons $i$ and $j$ belong to, the value of the synaptic coupling $J_{ij}$ is drawn randomly from a distribution in which statistics depend on the pre- and postsynaptic population $q$ and $p$. The full connectivity matrix $\mathbf{J}$ therefore has a

block structure, in the sense that all connections within the same block share *identical* statistics.

We examine two variants of this model class: (1) **independent** random connectivity [53]; (2) connectivity with **reciprocal motifs** [10, 82]. In each case, we examine two specific examples of distributions of synaptic strengths, *Gaussian*, and *Sparse* distributions.

**2.1.1 Independent random connectivity.**   For networks with independent random connectivity, the recurrent connections $J_{ij}$ are sampled independently for each $(i, j)$ pair from

$$\text{Prob}(J_{ij} = J) = f^{pq}(J), \tag{22}$$

where $f^{pq}$ denotes a probability density function, and $q, p$ are the pre- and post-synaptic populations. Separating the mean and random components, for an arbitrary distribution Eq (22) can be re-expressed as

$$J_{ij} = \bar{J}_{pq} + z_{ij}. \tag{23}$$

Here $\bar{J}_{pq}$ is the mean value of the connections from population $q$ to population $p$, and $z_{ij}$ is the remaining zero-mean random part of each connection. Defining $\bar{\mathbf{J}}$ as the $N \times N$ deterministic matrix consisting of mean values, and $\mathbf{Z}$ as the noise matrix consisting of the random parts $z_{ij}$, the connectivity matrix $\mathbf{J}$ can be written as

$$\mathbf{J} = \bar{\mathbf{J}} + \mathbf{Z}. \tag{24}$$

The matrix $\bar{\mathbf{J}}$ is of size $N \times N$ and consists of $P^2$ blocks with identical values within each block. The rank of $\bar{\mathbf{J}}$ is therefore at most $P$ [40]. In contrast, the noise matrix $\mathbf{Z}$ is in general of rank $N$. The full connectivity matrix $\mathbf{J}$ can then be interpreted as a rank-$P$ deterministic matrix perturbed by the random matrix $\mathbf{Z}$ with block-dependent statistics.

In the case of Gaussian connectivity, connections from population $q$ to population $p$ are sampled independently from a Gaussian distribution.

$$f^{pq}(\mathbf{J}) = \mathcal{N}(\bar{J}_{pq}, \sigma^2_{z_{pq}}), \tag{25}$$

with variances

$$\sigma^2_{z_{pq}} = \frac{g^2_{pq}}{N}. \tag{26}$$

The noise matrix $\mathbf{Z}$ therefore has *block-structured variances* $g^2_{pq}/N$ that we specify by a $P \times P$ matrix $\mathbf{G}_m$:

$$\mathbf{G}_m = \begin{bmatrix} g^2_{11}/N & \cdots & g^2_{1P}/N \\ \vdots & & \vdots \\ g^2_{P1}/N & \cdots & g^2_{PP}/N \end{bmatrix}. \tag{27}$$

In the case of sparse connectivity, $J^{pq}_{ij}$ is a Bernoulli random variable. The connectivity weights $J^{pq}_{ij}$ from population $q$ to population $p$ are non-zero with probability $c_{pq}$ and zero otherwise. All non-zero connection weights within a block take the same value $A_{pq}$, so that

analogously to Eq (22), the sparse connectivity is defined as

$$
f^{pq}(J_{ij} = J) = \begin{cases} c_{pq} & \text{for } J = A_{pq}, \\ 1 - c_{pq} & \text{for } J = 0. \end{cases} \tag{28}
$$

The mean connectivity weight between populations $p$, $q$ is then

$$
\bar{J}_{pq} = c_{pq} A_{pq}. \tag{29}
$$

and the variance of the remaining random part $z_{ij}$ is

$$
\sigma^2_{z_{pq}} = [z^2_{ij}]_{i \in N_p, j \in N_q} = (1 - c_{pq}) c_{pq} A^2_{pq}, \tag{30}
$$

to simplify the parameters in sparse networks, we assume that $A_{pq}$ depend only on presynaptic population $q$, and that the connection probability $c_{pq}$ is a homogeneous network parameter independent of $p$, $q$ that we denote by $c$.

**2.1.2 Reciprocal connectivity motifs.** To go beyond independent connectivity, we consider pairwise motifs, i. e. correlations between reciprocal pairs of weights $J_{ij}$ and $J_{ji}$. We quantify this correlation using the normalized covariance $\eta_{ij}$ defined as

$$
\eta_{ij} = \frac{[(J_{ij} - [J_{ij}])(J_{ji} - [J_{ji}])]}{\sqrt{[(J_{ij} - [J_{ij}])^2][(J_{ji} - [J_{ji}])^2]}}, \tag{31}
$$

where $[\cdot]$ denotes the average over the full connectivity distribution. Reciprocal connections are fully independent when $\eta_{ij} = 0$ for all $i, j$, fully symmetric when $\eta_{ij} = 1$ and fully anti-symmetric when $\eta_{ij} = -1$.

Our key assumption is that the statistics of connectivity are block-like, implying that all pairs of connections between populations $p$, $q$ share the same correlation coefficient $\eta_{pq}$, so that the statistics are defined by a $P \times P$ reciprocal correlation matrix $\boldsymbol{\eta}_m$

$$
\eta_m = \begin{bmatrix} \eta_{11} & \cdots & \eta_{1P} \\ \vdots & & \vdots \\ \eta_{P1} & \cdots & \eta_{PP} \end{bmatrix}, \tag{32}
$$

where, by definition $\eta_{pq} = \eta_{qp}$.

For Gaussian statistics, we generate connectivity matrices with a specified set of $\eta_{pq}$ in the following manner. We first generate an $N \times N$ matrix $\mathbf{Y}'$ with entries independently sampled from the normal distribution $\mathcal{N}(0, 1)$. Then, in order to generate a matrix $\mathbf{Y}$ with reciprocal correlations $\eta_{pq}$, we form a linear combination of $\mathbf{Y}'$ and its transpose $\mathbf{Y}'^\top$. Specifically, we set

$$
y_{ij} = \underbrace{\gamma_{pq} y'_{ij}}_{\mathbf{Y}'} + \underbrace{\sqrt{1 - \gamma^2_{pq}} y'_{ji}}_{\mathbf{Y}'^\top} \tag{33}
$$

with $\gamma_{pq} = \sqrt{(1 - \sqrt{1 - \eta^2_{pq}})/2}$ for $\eta_{pq} > 0$, and $-\sqrt{(1 - \sqrt{1 - \eta^2_{pq}})/2}$ for $\eta_{pq} < 0$. Finally, we scale each block by $g_{pq}/\sqrt{N}$ to obtain the random connectivity component $\mathbf{Z}$, which is added to the mean connectivity component $\bar{\mathbf{J}}$ to finally obtain the full connectivity matrix $\mathbf{J}$.

For *sparse* networks, we first generate a connectivity matrix without reciprocal correlations. We then consider the upper triangle of this matrix, randomly select a fraction $\rho_{pq}$ of the non-

zero connections $J_{ij}$ with value $A_q$ and set their reciprocal connectivity weights $J_{ji}$ to have a non-zero weight $A_p$. For the remaining $1 - \rho_{pq}$ fraction of non-zero connections in the upper triangle, we set the reciprocal connectivity weights to zero. The corresponding cell-type dependent reciprocal correlations for the multi-population sparse connectivity are then

$$\eta_{pq} = \frac{A_p A_q (\rho_{pq} - c)}{|A_p A_q|(1 - c)}, \quad p, q = 1 \ldots P, \tag{34}$$

where $c$ is the homogeneous connection probability (Table 1).

**2.1.3 Excitatory-inhibitory networks.** In this work, we specifically focus on excitatory-inhibitory networks composed of $P = 2$ populations, one excitatory and one inhibitory, with respectively $N_E$ and $N_I$ neurons. We denote the two populations by indices $E$ and $I$, so that there are four types of connections: $EE$, $EI$, $IE$ and $II$. Based on the usual anatomical estimates for neocortex, we choose $N_E = 0.8N$, $N_I = 0.2N$, and further define $\alpha_E = N_E/N$, $\alpha_I = N_I/N$, as the fractions of excitatory and inhibitory neurons.

For Gaussian networks, we enforce Dale's law only on the mean, i. e. we set $\bar{J}_{EE}$ and $\bar{J}_{IE}$ to be positive, while $\bar{J}_{EI}$ and $\bar{J}_{II}$ are negative. The $N \times N$ mean connectivity matrix $\bar{\mathbf{J}}$ is therefore in general rank-two. To further simplify the setting, we follow [43], and consider networks where the mean weights of all excitatory connections, and respectively all inhibitory connections, are equal and set by parameters $J_E$ and $J_I$:

$$\bar{J}_{EE} = \bar{J}_{IE} = J_E/N_E \tag{35}$$

$$\bar{J}_{EI} = \bar{J}_{II} = -J_I/N_I. \tag{36}$$

Under these additional assumptions, the entries in the first $N_E$ columns of the mean connectivity matrix $\bar{\mathbf{J}}$ have the same positive weight $J_E/N_E$, and the entries in the following $N_I$ columns have the same negative weight $-J_I/N_I$, so that $\bar{\mathbf{J}}$ becomes rank one.

We however allow the variances $g_{pq}^2/N$ and reciprocal correlations $\eta_{pq}$ to depend on both the pre- and post-synaptic population, so that the corresponding parameters form $2 \times 2$ matrices

$$\mathbf{G}_m = \begin{bmatrix} g_{EE}^2/N & g_{EI}^2/N \\ g_{IE}^2/N & g_{II}^2/N \end{bmatrix}, \quad \boldsymbol{\eta}_m = \begin{bmatrix} \eta_{EE} & \eta_{EI} \\ \eta_{IE} & \eta_{II} \end{bmatrix}, \tag{37}$$

where $\eta_{EI} = \eta_{IE}$.

For *sparse* excitatory-inhibitory networks, all non-zero excitatory (resp. inhibitory) synaptic weights are equal and positive, $A_E > 0$ (resp. $A_I < 0$). From Eqs (29) and (30) the mean and the variance of the synaptic weights in the sparse network can be matched to the parameters of the Gaussian model [83]:

$$\frac{J_E}{N_E} = cA_E, \quad g_{EE}^2/N = g_{IE}^2/N = A_E^2 c(1 - c),$$

$$-\frac{J_I}{N_I} = cA_I, \quad g_{EI}^2/N = g_{II}^2/N = A_I^2 c(1 - c), \tag{38}$$

In particular, for the sparse networks with pairwise reciprocal motifs, on top of the matching means and variances, the cell-type dependent reciprocal correlations satisfy (Eq (34))

$$\eta_{EE} = \frac{\rho_{EE} - c}{1 - c}, \quad \eta_{EI} = -\frac{\rho_{EI} - c}{1 - c}, \quad \eta_{II} = \frac{\rho_{II} - c}{1 - c}. \tag{39}$$

## 2.2 Globally-defined connectivity: Gaussian-mixture low-rank networks

In this section, we introduce a second broad class of connectivity models, Gaussian-mixture low-rank networks [31, 33], in which the connectivity matrix is generated from a global statistics of vectors over neurons.

Low-rank networks are a class of recurrent neural networks in which the connectivity matrix $\mathbf{J}$ is restricted to be of rank $R$, assumed to be much smaller than the number of neurons $N$. Such a connectivity matrix can be expressed as a sum of $R$ unit rank terms

$$\mathbf{J} = \frac{1}{N} \sum_{r=1}^{R} \mathbf{m}^{(r)} \mathbf{n}^{(r)\top}. \tag{40}$$

We refer to $\mathbf{n}^{(r)} = \{n_i^{(r)}\}_{i=1\ldots N}$ and $\mathbf{m}^{(r)} = \{m_i^{(r)}\}_{i=1\ldots N}$ as the $r$-th left and right *connectivity vectors*. The $2R$ connectivity vectors together fully specify the connectivity matrix. Each neuron $i$ is then characterized by its set of $2R$ entries $(m_i^{(1)}, n_i^{(1)}, \ldots, m_i^{(R)}, n_i^{(R)})$ on these vectors. For unit-rank networks, the main focus of this study, the connectivity matrix is simply given by the outer product of a pair of connectivity vectors $\mathbf{m}$ and $\mathbf{n}$:

$$\mathbf{J}_{R1} = \frac{1}{N} \mathbf{m} \mathbf{n}^\top. \tag{41}$$

Gaussian-mixture low-rank networks are a subset of the class of low-rank networks, for which the entries of the connectivity vectors are drawn independently for each neuron from a mixture of Gaussians distribution [31]. Specifically, a fraction $\alpha_p$ of neurons is assigned to a population $p$, and within each population, the entries on the connectivity vectors are generated from a given $2R$-dimensional Gaussian distribution. For a unit-rank network, for a neuron $i$ in the population $p$, the connectivity parameters $(m_i, n_i)$ are generated from a bi-variate Gaussian distribution with mean $(\bar{m}^p, \bar{n}^p)$, variance $(\sigma_{m^p}^2, \sigma_{n^p}^2)$ and covariance $\sigma_{nm}^p$.

For any unit-rank matrix of the form in Eq (41), the only potentially non-zero eigenvalue is given by $\lambda = \mathbf{n}^\mathsf{T} \mathbf{m}/N$, and the corresponding right (resp. left) eigenvector is $\mathbf{m}$ (resp. $\mathbf{n}$). For a Gaussian-mixture model, in the large $N$ limit this eigenvalue becomes

$$\lambda = \sum_{p=1}^{P} \alpha_p (\bar{m}^p \bar{n}^p + \sigma_{nm}^p) \tag{42}$$

Starting from a Gaussian-mixture low-rank model in which the connectivity is globally defined, and the assumptions that $m_i$, $n_i$ are drawn independently between neurons from a multi-variate Gaussian distribution is satisfied, it is straightforward to compute the resulting local statistics of the connectivity, i. e. the mean $\bar{J}_{pq}$, variance $\sigma_{z_{pq}}^2$ (Methods Sec. 2.1.2) and

reciprocal correlation $\eta_{pq}$ (Methods Sec. 2.1.1) as:

$$\bar{J}_{pq} = \frac{1}{N}\bar{m}^p\bar{n}^q,$$

$$\sigma_{z_{pq}}^2 = \frac{1}{N^2}\left(\sigma_{m^p}^2(\bar{n}^q)^2 + (\bar{m}^p)^2\sigma_{n^q}^2 + \sigma_{m^p}^2\sigma_{n^q}^2\right), \tag{43}$$

$$\eta_{pq} = \frac{(\sigma_{nm}^p\bar{m}^q\bar{n}^q + \sigma_{nm}^q\bar{m}^p\bar{n}^p + \sigma_{nm}^p\sigma_{nm}^q)}{\sqrt{(\sigma_{m^p}^2(\bar{n}^q)^2 + (\bar{m}^p)^2\sigma_{n^q}^2 + \sigma_{m^p}^2\sigma_{n^q}^2)(\sigma_{m^q}^2(\bar{n}^p)^2 + (\bar{m}^q)^2\sigma_{n^p}^2 + \sigma_{m^q}^2\sigma_{n^p}^2)}}.$$

in the large network limit. The expression for the local statistics of network connectivity using rank-$R$ connectivity is in S5 Text.

## 2.3 Approximating locally-defined connectivity with Gaussian-mixture low-rank models

In this section, we describe our general approach for approximating an arbitrary connectivity matrix $\mathbf{J}$ with a rank-$R$ matrix $\mathbf{J}_R$. We then show that for $\mathbf{J}$ corresponding to locally-defined multi-population connectivity (Methods Sec. 2.1), the resulting approximation $\mathbf{J}_R$ in general obeys Gaussian-mixture low-rank statistics as defined in Methods Sec. 2.2.

The connectivity matrices $\mathbf{J}$ that we studied have randomly generated entries. Such matrices can be diagonalized almost surely (with probability 1) on complex numbers, as the set of non-diagonalizable matrices is of measure 0. The matrices we consider are however not symmetric and therefore non-normal [41], so that the left and right eigenvectors are in general not identical. Instead, they form a bi-orthogonal set [84]. Specifically, to approximate a full rank matrix $\mathbf{J}$ with a rank-$R$ matrix $\mathbf{J}_R$, we use truncated eigen-decomposition, which preserves the dominant eigenvalues. We start from the full eigen-decomposition of $\mathbf{J}$:

$$\mathbf{J} = \sum_{r=1}^{N}\lambda_r\mathbf{R}_r\mathbf{L}_r^T, \tag{44}$$

where $\lambda_r$ is the $r$-th eigenvalue of $\mathbf{J}$ (ordered by decreasing absolute value), while $\mathbf{R}_r$ and $\mathbf{L}_r$ are the corresponding right- and left-eigenvectors that obey

$$\mathbf{J}\mathbf{R}_r = \mathbf{R}_r\lambda_r \tag{45}$$

$$\mathbf{L}_r^{\mathsf{T}}\mathbf{J} = \lambda_r\mathbf{L}_r^{\mathsf{T}} \tag{46}$$

$$\mathbf{L}_r^T\mathbf{R}_{r'} = \delta_{rr'}. \tag{47}$$

In the following, we constrain the right eigenvectors $\mathbf{R}_r$ to be of unit norm, while the normalization of the left eigenvector is determined by Eq (47).

We obtain a rank-$R$ approximation $\mathbf{J}_R$ of $\mathbf{J}$ by keeping the first $R$ terms in Eq (44):

$$\mathbf{J}_R = \sum_{r=1}^{R}\lambda_r\mathbf{R}_r\mathbf{L}_r^T. \tag{48}$$

The $R$ non-trivial eigenvalues and eigenvectors of $\mathbf{J}_R$ therefore correspond to the first $R$ eigenvalues and eigenvectors of $\mathbf{J}$. We then set

$$\mathbf{m}^{(r)} = \sqrt{N}\mathbf{R}_r \tag{49}$$

$$\mathbf{n}^{(r)} = \sqrt{N}\lambda_r \mathbf{L}_r \tag{50}$$

to have the same normalization for $\mathbf{J}_R$ as in Eq (40).

To obtain a low-rank approximation for a connectivity matrix $\mathbf{J}$ generated from locally-defined statistics defined in Methods Sec. 2.1, we first determine its dominant eigenvalues and eigenvectors. Starting from Eq (24), this problem becomes equivalent to finding the dominant eigenvalues and eigenvectors of a low-rank matrix $\bar{\mathbf{J}}$ perturbed by a random matrix $\mathbf{Z}$ with block-like statistics. We compute the statistics of these eigenvalues and eigenvectors by combining the Matrix's Determinant Lemma, the Matrix Perturbation Theory and the Central Limit Theorem. Below we summarize this general approach before applying it to different specific cases in Methods Secs. 2.4, 2.5.

We focus on the case where $\bar{\mathbf{J}}$ is unit rank as in the simplified excitatory-inhibitory network introduced in Methods Sec. 2.1.3. In that case, the unique non-zero eigenvalue of $\bar{\mathbf{J}}$ is

$$\lambda_0 = J_E - J_I, \tag{51}$$

and the corresponding left and right eigenvectors are

$$
\begin{aligned}
\bar{\mathbf{L}} &= \left[\frac{\sqrt{N}J_E}{N_E(J_E - J_I)}, \dots, -\frac{\sqrt{N}J_I}{N_I(J_E - J_I)}, \dots\right]^{\mathsf{T}}, \\
\bar{\mathbf{R}} &= \left[\frac{1}{\sqrt{N}}, \dots, \frac{1}{\sqrt{N}}, \dots\right]^{\mathsf{T}}.
\end{aligned} \tag{52}
$$

$\bar{\mathbf{J}}$ can then be rewritten as

$$\bar{\mathbf{J}} = \frac{1}{N}\bar{\mathbf{m}}\bar{\mathbf{n}}^{\mathsf{T}}, \tag{53}$$

where the structure vectors $\bar{\mathbf{m}}$ and $\bar{\mathbf{n}}$ are uniquely defined by rescaling the left and right eigenvectors $\bar{\mathbf{L}}$, $\bar{\mathbf{R}}$ of $\bar{\mathbf{J}}$ (Eq (52)) as in Eq (49), so that

$$\bar{m}_i = 1, \quad i = 1\dots N \tag{54}$$

$$\bar{n}_i = \bar{n}^E = \frac{N}{N_E}J_E \quad i \in N_E \tag{55}$$

$$\bar{n}_i = \bar{n}^I = -\frac{N}{N_I}J_I \quad i \in N_I, \tag{56}$$

The full connectivity matrix $\mathbf{J}$ can be then expressed as

$$\mathbf{J} = \bar{\mathbf{m}}\bar{\mathbf{n}}^{\mathsf{T}}/N + \mathbf{Z}. \tag{57}$$

*Eigenvalues.* For a random matrix $\mathbf{Z}$ with independently distributed elements, the eigenvalues are distributed on a disk of radius $r_g$ centred at the origin in the complex plane [11, 44, 85]. Correlations between elements in general modify the shape of this continuous spectrum [18, 86]. In contrast, adding a low-rank component typically induces isolated eigenvalues outside the continuous part of the spectrum [44, 45]. To obtain a low-rank approximation of the full matrix, we focus on determining these outliers when they exist.

All eigenvalues $\lambda$ of $\mathbf{J}$ satisfy the characteristic equation

$$\det(\mathbf{J} - \mathbf{I}\lambda) = 0. \tag{58}$$

To determine the outlying eigenvalues of a random connectivity with low-rank structure, we apply the matrix determinant lemma to the l. h. s. of the characteristic equation [32]:

$$\det\left(\frac{1}{N}\bar{\mathbf{m}}\bar{\mathbf{n}}^{\mathsf{T}} + \mathbf{Z} - \mathbf{I}\lambda\right) = \left(1 + \frac{1}{N}\bar{\mathbf{n}}^{\mathsf{T}}(\mathbf{Z} - \mathbf{I}\lambda)^{-1}\bar{\mathbf{m}}\right)\det(\mathbf{Z} - \mathbf{I}\lambda). \tag{59}$$

As outliers by definition cannot be eigenvalues of $\mathbf{Z}$, they correspond to zeros of the first term in the r. h. s., and therefore satisfy:

$$\lambda = \frac{1}{N}\bar{\mathbf{n}}^{\mathsf{T}}(\mathbf{I} - \mathbf{Z}/\lambda)^{-1}\bar{\mathbf{m}}, \tag{60}$$

where $\mathbf{I}$ is the identity matrix. We assume that the maximal eigenvalue of $\mathbf{Z}$ is smaller than the outlying eigenvalues $\lambda$ we aim to determine, the upper bound on this maximal eigenvalue is provided by the spectral norm of the $\mathbf{Z}$ matrix [87]. As a result, we can expand $(\mathbf{I} - \mathbf{Z}/\lambda)^{-1}$ in series, and further get [32]

$$\lambda = \sum_{k=0}^{\infty}\frac{\theta_k}{\lambda^k}, \tag{61}$$

with

$$\theta_k = \frac{1}{N}\bar{\mathbf{n}}^{\mathsf{T}}\mathbf{Z}^k\bar{\mathbf{m}}. \tag{62}$$

Here $\theta_0$ corresponds to the eigenvalue $\lambda_0$ of $\bar{\mathbf{J}}$ (Eq (51)), and the higher order terms specify how this eigenvalue is modified by the random part of the connectivity. Truncating Eq (61) at a given order, and averaging over $\mathbf{Z}$ yields a polynomial equation for the mean eigenvalues of $\mathbf{J}$. In Methods Sec. 2.4, we exploit this equation to determine the effects of different cell-type specific random connectivity $\mathbf{Z}$ on the outlying eigenvalues.

Note that within first-order perturbation theory, the eigenvalues are given by $\lambda = \lambda_0 + \Delta\lambda$ with

$$\Delta\lambda = \bar{\mathbf{L}}^{\mathsf{T}}\mathbf{Z}\bar{\mathbf{R}} = \frac{1}{N\lambda_0}\bar{\mathbf{n}}^{\mathsf{T}}\mathbf{Z}\bar{\mathbf{m}}. \tag{63}$$

*Eigenvectors.* To determine the eigenvectors corresponding to the outlying eigenvalue of $\mathbf{J}$, we treat it as $\bar{\mathbf{J}}$ perturbed by $\mathbf{Z}$ (Eq (24)). Matrix perturbation theory then states that, at first order, the right- and left-eigenvectors $\mathbf{R}$ and $\mathbf{L}$ of $\mathbf{J}$ corresponding to the outlying eigenvalue $\lambda$ are given by [47]:

$$\mathbf{R} = \bar{\mathbf{R}} + \Delta\mathbf{R} \tag{64}$$

$$\mathbf{L} = \bar{\mathbf{L}} + \Delta\mathbf{L} \tag{65}$$

where $\bar{\mathbf{R}}$ and $\bar{\mathbf{L}}$ are the right- and left-eigenvectors of $\bar{\mathbf{J}}$ defined in Eq (52), and

$$\Delta\mathbf{R} = \frac{1}{\lambda_0}\mathbf{Z}\bar{\mathbf{R}}$$

$$\Delta\mathbf{L}^\mathsf{T} = \frac{1}{\lambda_0}\bar{\mathbf{L}}^\mathsf{T}\mathbf{Z}.$$

(66)

Using the normalization in Eq (49), we then get

$$\mathbf{m} = \sqrt{N}\mathbf{R} = \bar{\mathbf{m}} + \Delta\mathbf{m},$$

$$\mathbf{n}^\mathsf{T} = \lambda\sqrt{N}\mathbf{L}^\mathsf{T} = \bar{\mathbf{n}}^\mathsf{T} + \Delta\mathbf{n}^\mathsf{T},$$

(67)

with constant entries on $\bar{\mathbf{m}}$ and $\bar{\mathbf{n}}$ defined in Eqs (54) and (56) and

$$\Delta\mathbf{m} = \frac{1}{\lambda_0}\mathbf{Z}\bar{\mathbf{m}}$$

$$\Delta\mathbf{n}^\mathsf{T} = \frac{1}{\lambda_0}\bar{\mathbf{n}}^\mathsf{T}\mathbf{Z}.$$

(68)

where we approximated $\lambda$ at first order by $\lambda_0$.

***Statistics of Eigenvector entries.*** While $\bar{\mathbf{m}}$ and $\bar{\mathbf{n}}$ are deterministic vectors, the perturbations $\Delta\bar{\mathbf{m}}$ and $\Delta\bar{\mathbf{n}}$ are random variables obtained by multiplying $\bar{\mathbf{m}}$ and $\bar{\mathbf{n}}$ with the random matrix $\mathbf{Z}$ (Eq (68)). We therefore next consider the statistics of the elements $m_i$ and $n_i$ of $\mathbf{m}$ and $\mathbf{n}$ defined in Eq (67).

Since the elements of $\mathbf{Z}$ have zero mean, the mean values of $m_i$ and $n_i$ are given by $\bar{m}_i$ and $\bar{n}_i$ defined in Eqs (54) and (56). The mean value of $n_i$, but not $m_i$, therefore depends on whether the neuron $i$ belongs to the excitatory or inhibitory population. Taking into account that $\mathbf{Z}$ has block-like statistics, we split the matrix product in Eq (68) into the sum of items corresponding to excitatory and inhibitory pre-synaptic neurons. Using Eqs (54) and (56), $\Delta m_i$ and $\Delta n_i$ can be written as

$$\Delta m_i = \frac{1}{\lambda_0}\sum_{j\in N_E}z_{ij} + \frac{1}{\lambda_0}\sum_{j\in N_I}z_{ij}$$

$$\Delta n_i = \frac{1}{\lambda_0}\sum_{j\in N_E}\bar{n}^E z_{ji} + \frac{1}{\lambda_0}\sum_{j\in N_I}\bar{n}^I z_{ji},$$

(69)

We next take the limit $N_E, N_I \to \infty$, and apply the central limit theorem, which states that each sum converges to a Gaussian random variable, so that we have

$$\Delta m_i^p \sim \frac{1}{\lambda_0}\sum_{q=E,I}\sqrt{N_q}\mathcal{N}(0,\sigma_{z_{pq}}^2)$$

$$\Delta n_i^p \sim \frac{1}{\lambda_0}\sum_{q=E,I}\sqrt{N_q}\bar{n}^q\mathcal{N}(0,\sigma_{z_{qp}}^2).$$

(70)

where $p \in E, I$ is the population the neuron $i$ belongs to, and $\sigma_{z_{pq}}^2$, $\sigma_{z_{qp}}^2$ are the variance of $z_{ij}, z_{ji}$ respectively, for $i, j$ in populations $p, q$. The perturbations $\Delta m_i$ and $\Delta n_i$. therefore converge to

Gaussian random variables of zero mean and variances $\sigma_{m^p}^2$ and $\sigma_{n^p}^2$ given by:

$$
\begin{aligned}
\sigma_{m^p}^2 &= \sum_{q=E,I} \frac{N_q \sigma_{z_{pq}}^2}{\lambda_0^2} \\
\sigma_{n^p}^2 &= \sum_{q=E,I} \frac{N_q (\bar{n}^q)^2 \sigma_{z_{qp}}^2}{\lambda_0^2}.
\end{aligned}
\tag{71}
$$

In order to guarantee stability in the large network limit, we assume that the variance of the local random synaptic weights is satisfying $\sigma_{z_{pq}}^2 = \mathcal{O}(1/N)$. Consequently, we have $\sigma_{m^p}^2 = \mathcal{O}(1)$ and $\sigma_{n^p}^2 = \mathcal{O}(1)$ (Eqs (70) and (71)).

The population covariance $\sigma_{nm}^p$ between elements $m_i$ and $n_i$ with $i$ belonging to population $p$ can furthermore be written as

$$
\sigma_{nm}^p = \frac{1}{N_p} \Delta \mathbf{n}^{p\top} \Delta \mathbf{m}^p = \frac{1}{N_p \lambda_0^2} \sum_{s,q=E,I} \left( \bar{n}^s \sum_{i \in N_s} \sum_{j \in N_q} \sum_{k \in N_p} z_{ik} z_{kj} \right),
\tag{72}
$$

while the overall covariance $\sigma_{nm}$ between all $m_i$ and $n_i$ reads

$$
\sigma_{nm} = \sum_{p=E,I} \alpha_p \sigma_{nm}^p.
\tag{73}
$$

Altogether, $m_i$ and $n_i$ determined by perturbation theory therefore follow Gaussian-mixture statistics, where the mean and variance depend on whether the neuron $i$ belongs to the excitatory or inhibitory population.

***Comparison with simulations.*** The theoretical predictions for eigenvalues obtained from Eqs (61) and (62) can be verified by comparing them with the eigenvalue outliers computed by direct eigen-decomposition of the full matrix $\mathbf{J}$. We compute the average and standard deviation of eigenvalue outliers over 30 realizations of $\mathbf{J}$.

The predictions of perturbation theory for eigenvectors given by Eq (66) can also be verified by direct eigen-decomposition, but to compare individual entries, an appropriate normalization is required [47]. Indeed, perturbation theory assumes that $\bar{\mathbf{R}}$ is normalized and $\bar{\mathbf{L}}$ satisfies $\bar{\mathbf{L}}^\top \bar{\mathbf{R}} = 1$ (Eq (52)), but the perturbed eigenvectors in Eq (64) do not obey the same normalization. We therefore first use numerical eigen-decomposition to get the right- and left-eigenvector $\hat{\mathbf{R}}$ and $\hat{\mathbf{L}}$ of $\mathbf{J}$. We then normalize $\hat{\mathbf{R}}$ to 1, and $\hat{\mathbf{L}}$ so that $\hat{\mathbf{L}}^\top \hat{\mathbf{R}} = 1$. To compare $\hat{\mathbf{L}}, \ \hat{\mathbf{R}}$ with perturbation theory, we then normalize $\hat{\mathbf{L}}, \ \hat{\mathbf{R}}$ as

$$
\begin{aligned}
\mathbf{R} &= \left( \frac{\hat{\mathbf{L}}^\top \bar{\mathbf{R}}}{\bar{\mathbf{L}}^\top \hat{\mathbf{R}}} \right)^{1/2} \hat{\mathbf{R}}, \\
\mathbf{L} &= \left( \frac{\hat{\mathbf{R}}^\top \bar{\mathbf{L}}}{\bar{\mathbf{R}}^\top \hat{\mathbf{L}}} \right)^{1/2} \hat{\mathbf{L}},
\end{aligned}
\tag{74}
$$

the eigenvectors $\mathbf{L}, \mathbf{R}$ after normalization have the same statistics as $(\bar{\mathbf{L}} + \Delta \mathbf{L})$, $(\bar{\mathbf{R}} + \Delta \mathbf{R})$ (Eqs (62) and (64)).

## 2.4 Eigenvalues

Here, we apply Eqs (61) and (62) to determine the mean and variance of outlying eigenvalues for different forms of local connectivity statistics.

**2.4.1 Independent random connectivity.**   In the case of independent random connectivity, the elements of $\mathbf{Z}$ are zero-mean, independently distributed and uncorrelated with $\bar{\mathbf{m}}$ and $\bar{\mathbf{n}}$. Averaging Eq (62) over $\mathbf{Z}$ then leads to [32]:

$$[\theta_k] = \left[\frac{1}{N}\bar{\mathbf{n}}^\top \mathbf{Z}^k \bar{\mathbf{m}}\right] \tag{75}$$

$$= 0, \tag{76}$$

in the limit $N \to \infty$ for all $k > 0$, and therefore the mean eigenvalue $[\lambda]$ of $\mathbf{J}$ is given by the eigenvalue $\lambda_0$ of $\bar{\mathbf{J}}$. For Gaussian random connectivity, we have

$$[\lambda] = J_E - J_I, \tag{77}$$

and for sparse connectivity

$$[\lambda] = c(N_E A_E + N_I A_I). \tag{78}$$

The variance $\sigma_\lambda^2$ of $\lambda$ can be computed by keeping only the linear term in Eq (61), which leads to Eq (63) under the assumption that $\lambda \approx \lambda_0$. Applying the central limit theorem then yields

$$\sigma_\lambda^2 = \frac{1}{\lambda_0^2}\left(J_E^2\left(\sigma_{z_{EE}}^2 + \frac{N_I}{N_E}\sigma_{z_{EI}}^2\right) + J_I^2\left(\frac{N_E}{N_I}\sigma_{z_{IE}}^2 + \sigma_{z_{II}}^2\right)\right). \tag{79}$$

Given the stability guarantee $\sigma_{z_{pq}}^2 = \mathcal{O}(1/N)$, the asymptotic variance of the first-order perturbation of the eigenvalue $\sigma_\lambda^2$ becomes zero in the limit of large network $N \to \infty$.

For independent Gaussian random connectivity, we substitute $\sigma_{z_{pq}}^2$ using Gaussian variance parameters (Eqs (25) and (27))

$$\sigma_\lambda^2 = \frac{1}{N\lambda_0^2}\left(J_E^2\left(g_{EE}^2 + \frac{N_I}{N_E}g_{EI}^2\right) + J_I^2\left(\frac{N_E}{N_I}g_{IE}^2 + g_{II}^2\right)\right). \tag{80}$$

For independent sparse random connectivity, we replace $\sigma_{z_{pq}}^2$ and $J_p$ with the variances and means of the sparse model given in Eqs (29) and (30) and get

$$\sigma_\lambda^2 = \frac{1}{\lambda_0^2}\left(A_E^2 N_E + A_I^2 N_I\right)^2 (1-c)c^3. \tag{81}$$

**2.4.2 Reciprocal motifs.**   In the case of connectivity with reciprocal correlations, $z_{ij}$ and $z_{ji}$ are correlated, so that the average $[\theta_k]$ over $\mathbf{Z}$ in Eq (62) is non-zero for even $k$. Here we compute $[\theta_k]$ for $k = 2$ and truncate Eq (61) at second order to get a third-order polynomial equation for the mean eigenvalue:

$$f(\lambda) = \lambda^3 - (\lambda_0\lambda^2 + [\theta_1]\lambda + [\theta_2]) = 0. \tag{82}$$

For Gaussian connectivity we have $\theta_0 = \lambda_0 = J_E - J_I$, and $\theta_1$ is given by

$$\theta_1 = \bar{\mathbf{n}}^\top \mathbf{Z}\bar{\mathbf{m}}/N = \sum_{p,q=E,I}\bar{n}^p\sum_{i\in N_p, j\in N_q} z_{ij}\bar{m}^q/N, \tag{83}$$

so that $[\theta_1] = 0$.

The next term $\theta_2$ is

$$\theta_2 = \bar{\mathbf{n}}^{\mathsf{T}} \mathbf{Z}^2 \bar{\mathbf{m}} / N = \sum_{p,q=E,I} \bar{n}^p \left( \sum_{i \in N_p} \sum_{j \in N_q} \sum_{k=1}^{N} z_{ik} z_{kj} \right) \bar{m}^q / N. \tag{84}$$

Given the reciprocal correlations defined in Eq (31), only items with $i = j$ in $\theta_2$ are non-zero after averaging over Gaussian realizations, so that

$$[\theta_2] = [\bar{\mathbf{n}}^{\mathsf{T}} \mathbf{Z}^2 \bar{\mathbf{m}}] / N = \sum_{p=E,I} \bar{n}^p [\sum_{i \in N_p} \sum_{k=1}^{N} z_{ik} z_{ki}] \bar{m}^p / N. \tag{85}$$

We then write

$$[\sum_{k=1}^{N} z_{ik} z_{ki}] = \alpha_E g_{EE}^2 \eta_{EE} + \alpha_I g_{EI} g_{IE} \eta_{EI}, \ i \in N_E$$

$$[\sum_{k=1}^{N} z_{ik} z_{ki}] = \alpha_E g_{IE} g_{EI} \eta_{EI} + \alpha_I g_{II}^2 \eta_{II}, \ i \in N_I. \tag{86}$$

and substitute Eq (86) and $\bar{n}^p, \ \bar{m}^p$ into Eq (85) to obtain

$$[\theta_2] = J_E (\alpha_E g_{EE}^2 \eta_{EE} + \alpha_I g_{EI} g_{IE} \eta_{EI})$$

$$- J_I (\alpha_E g_{IE} g_{EI} \eta_{EI} + \alpha_I g_{II}^2 \eta_{II}). \tag{87}$$

For sparse connectivity with reciprocal motifs, the correlations can be written as

$$[\sum_{k=1}^{N} z_{ik} z_{ki} = c A_E^2 N_E (\rho_{EE} - c) - c A_E A_I N_I (c - \rho_{EI}), \ i \in N_E$$

$$[\sum_{k=1}^{N} z_{ik} z_{ki}] = -c A_E A_I N_E (c - \rho_{EI}) + c A_I^2 N_I (\rho_{II} - c), \ i \in N_I. \tag{88}$$

Then, combining Eq (38) and Eqs (54) and (56), the second-order coefficient $[\theta_2]$ for the sparse network is

$$[\theta_2] = A_E^3 c^2 N_E^2 (\rho_{EE} - c) + A_E^2 A_I c^2 N_E N_I (\rho_{EI} - c)$$

$$+ A_E A_I^2 c^2 N_I N_E (\rho_{EI} - c) + A_I^3 c^2 N_I^2 (\rho_{II} - c). \tag{89}$$

Using Eqs (38) and (39), it can be seen that Eq (89) is equivalent to Eq (87).

## 2.5 Eigenvectors

Here we apply Eqs (71) and (72) to determine the variances and covariances of eigenvector entries obtained from perturbation theory for different forms of local connectivity statistics.

**2.5.1 Independent random connectivity.** In the case of independent random connectivity, because $z_{ik}$ and $z_{kj}$ are not correlated in Eq (72), the covariances $\sigma_{nm}^p$ between the eigenvector entries are zero. For independent Gaussian connectivity, introducing Eq (26) into Eq (71)

the variances of eigenvector entries can be written as

$$
\begin{aligned}
\sigma^2_{m^E} &= \frac{1}{\lambda_0^2}\left(\alpha_E g_{EE}^2 + \alpha_I g_{EI}^2\right), & \sigma^2_{m^I} &= \frac{1}{\lambda_0^2}\left(\alpha_E g_{IE}^2 + \alpha_I g_{II}^2\right), \\
\sigma^2_{n^E} &= \frac{1}{\lambda_0^2}\left(\frac{1}{\alpha_E}J_E^2 g_{EE}^2 + \frac{1}{\alpha_I}J_I^2 g_{IE}^2\right), & \sigma^2_{n^I} &= \frac{1}{\lambda_0^2}\left(\frac{1}{\alpha_E}J_E^2 g_{EI}^2 + \frac{1}{\alpha_I}J_I^2 g_{II}^2\right).
\end{aligned}
\tag{90}
$$

For independent sparse connectivity, substituting Eq (30) into Eq (71), leads to

$$
\begin{aligned}
\sigma^2_{m^E} &= \sigma^2_{m^I} = \frac{1}{\lambda_0^2}\left(N_E A_E^2 c(1-c) + N_I A_I^2 c(1-c)\right), \\
\sigma^2_{n^E} &= \frac{1}{\lambda_0^2}\left(A_E^2 N_E + A_I^2 N_I\right)A_E^2 N^2 c^3 (1-c), \\
\sigma^2_{n^I} &= \frac{1}{\lambda_0^2}\left(A_E^2 N_E + A_I^2 N_I\right)A_I^2 N^2 c^3 (1-c).
\end{aligned}
\tag{91}
$$

**2.5.2 Reciprocal motifs.** In the case of connectivity with reciprocal correlations, the variances of eigenvector entries are identical to the independent case.

As we have shown in Eq (72), noise correlation between the rank-one vectors arises from the correlation between pairwise random connectivity weights in the situation with reciprocal motifs, only items with $i = j$ (for $z_{ik}$, $z_{kj}$) in the same population $q$ are non-zero, so that we have

$$
\sigma_{nm} = \sum_{p=E,I} \alpha_p \sigma_{nm}^p
\tag{92}
$$

with

$$
\sigma_{nm}^p = \frac{1}{N_p \lambda_0^2} \sum_{q=E,I}\left(\bar{n}^q \sum_{i\in N_q}\sum_{k\in N_p} z_{ik} z_{ki} \bar{m}^q\right).
\tag{93}
$$

For Gaussian connectivity with reciprocal correlations, the covariances between entries on **Z** matched by population can be written as

$$
\begin{aligned}
\left[\sum_{i\in N_E}\sum_{k\in N_E} z_{ik} z_{ki}\right] &= N_E \alpha_E g_{EE}^2 \eta_{EE} \\
\left[\sum_{i\in N_E}\sum_{k\in N_I} z_{ik} z_{ki}\right] &= N_E \alpha_I g_{EI} g_{IE} \eta_{EI} \\
\left[\sum_{i\in N_I}\sum_{k\in N_E} z_{ik} z_{ki}\right] &= N_I \alpha_E g_{IE} g_{EI} \eta_{EI} \\
\left[\sum_{i\in N_I}\sum_{k\in N_I} z_{ik} z_{ki}\right] &= N_I \alpha_I g_{II}^2 \eta_{II}.
\end{aligned}
\tag{94}
$$

Combining Eq (94) and the mean rank-one connectivity loadings Eqs (54)–(56), we obtain the population covariances as

$$
\begin{aligned}
\sigma^E_{nm} &= \frac{1}{\lambda_0^2}\left(J_E g_{EE}^2 \eta_{EE} - J_I g_{EI} g_{IE} \eta_{EI}\right), \\[6pt]
\sigma^I_{nm} &= \frac{1}{\lambda_0^2}\left(J_E g_{EI} g_{IE} \eta_{EI} - J_I g_{II}^2 \eta_{II}\right).
\end{aligned}
\tag{95}
$$

We note that the large deviation of the dominant eigenvalue $\lambda$ in the network with reciprocal motifs also increases the nonlinearity of the vector perturbations. To account for this nonlinearity, we start from Eq (42) for $\lambda$ and get $\sigma_{nm} = \lambda - \sum_{p=E,I}\alpha_p \bar{m}^p \bar{n}^p = \lambda - \lambda_0$, then we compare with Eq (82) and get the approximation relationship

$$
\sigma_{nm} \approx \frac{\theta_2}{\lambda^2} = \frac{1}{N}\frac{(\bar{\mathbf{n}}^\top \mathbf{Z})(\mathbf{Z}\bar{\mathbf{m}})}{\lambda^2}.
\tag{96}
$$

Similarly, we substitute $\lambda$ for $\lambda_0$ in Eq (95) for the covariance of each population, and we have

$$
\begin{aligned}
\sigma^E_{nm} &= \frac{1}{\lambda^2}\left(J_E g_{EE}^2 \eta_{EE} - J_I g_{EI} g_{IE} \eta_{EI}\right), \\[6pt]
\sigma^I_{nm} &= \frac{1}{\lambda^2}\left(J_E g_{EI} g_{IE} \eta_{EI} - J_I g_{II}^2 \eta_{II}\right).
\end{aligned}
\tag{97}
$$

For sparse connectivity with reciprocal correlations, the calculations are similar, with entries of $\mathbf{Z}$ being Bernoulli-distributed

$$
\begin{aligned}
\Big[\sum_{i\in N_E}\sum_{k\in N_E} z_{ik}z_{ki}\Big] &= N_E^2 A_E^2 c(\rho_{EE} - c) \\[6pt]
\Big[\sum_{i\in N_E}\sum_{k\in N_I} z_{ik}z_{ki}\Big] &= -N_E N_I A_E A_I c(c - \rho_{EI}) \\[6pt]
\Big[\sum_{i\in N_I}\sum_{k\in N_E} z_{ik}z_{ki}\Big] &= -N_I N_E A_E A_I c(c - \rho_{EI}) \\[6pt]
\Big[\sum_{i\in N_I}\sum_{k\in N_I} z_{ik}z_{ki}\Big] &= N_I^2 A_I^2 c(\rho_{II} - c).
\end{aligned}
\tag{98}
$$

and we have the population covariance

$$
\begin{aligned}
\sigma^E_{nm} &= \frac{1}{\lambda^2} N A_E c^2 \left(A_E^2 N_E(\rho_{EE} - c) + A_I^2 N_I(\rho_{EI} - c)\right), \\[6pt]
\sigma^I_{nm} &= \frac{1}{\lambda^2} N A_I c^2 \left(A_E^2 N_E(\rho_{EI} - c) + A_I^2 N_I(\rho_{II} - c)\right).
\end{aligned}
\tag{99}
$$

Using Eqs (38) and (39), it can be seen that Eq (99) is equivalent to Eq (97).

## 2.6 Dynamics

In this section, we show how approximating locally-defined connectivity by a global low-rank structure allows us to analyse the emerging low-dimensional dynamics. We first summarize the mean-field theory (MFT) for Gaussian-mixture low-rank networks [31, 33]. We then apply

it to unit-rank connectivity obtained as an approximation of locally-defined connectivity. We finally compare the resulting description of the dynamics with an alternate mean-field approach for random connectivity consisting of a superposition of low-rank and full-rank random parts as in Eq (24) [30, 32].

Throughout this study, we consider recurrent networks of rate units with recurrent interactions defined by a connectivity matrix **J**. The dynamical activity of unit $i$ is represented by a variable $x_i(t)$, which we interpret as the total synaptic input current. The firing rate of unit $i$ is given by $r_i(t) = \phi(x_i(t))$ where $\phi(x) = 1 + \tanh(x - \theta)$ is a positive transfer function. We focus on networks without external inputs, so that the dynamics of synaptic input to neuron $i$ is given by

$$\dot{x}_i(t) = -x_i(t) + \sum_{j=1}^{N} J_{ij}\phi(x_i(t)). \tag{100}$$

In Figs 5–7, we compare the dynamics determined by direct simulations for a locally-defined connectivity matrix with a mean-field description obtained for a unit-rank approximation.

**2.6.1 Mean-field theory for Gaussian-mixture low-rank connectivity.** Here we review the mean-field theory for networks in which the connectivity matrix is exactly low-rank, with components of connectivity vectors moreover drawn from Gaussian-mixture distribution. Previous works have shown that in this case, the dynamics of the collective activity $\mathbf{x}(t) = \{x_i\}_{i=1\ldots N}$ are embedded in a linear subspace of dimension $R$ spanned by the connectivity vectors $\mathbf{m}^{(r)}$ [30–33]. Thus, $\mathbf{x}(t)$ can be expressed as

$$\mathbf{x}(t) = \sum_{r=1}^{R} \kappa_r(t)\mathbf{m}^{(r)}, \tag{101}$$

where $\kappa_r(t)$ for $r = 1\ldots R$ are collective latent variables that quantify the components of $\mathbf{x}(t)$ along the connectivity vectors $\mathbf{m}^{(r)}$. We assume that $\mathbf{m}^{(r)}$ are orthogonal to each other, so that $\kappa_r(t)$ can be expressed as

$$\kappa_r(t) = \frac{\mathbf{x}(t)^{\mathsf{T}}\mathbf{m}^{(r)}}{||\mathbf{m}^{(r)}||^2}. \tag{102}$$

For simplicity, here we moreover assume that the initial value of $\mathbf{x}(t)$ lies in the subspace spanned by the vectors $\mathbf{m}^{(r)}$. More generally, the initial state can be included as an additional input to the dynamics [31, 33].

For a unit rank connectivity $\mathbf{J} = \mathbf{m}\mathbf{n}^{\mathsf{T}}/N$, there is a single latent variable $\kappa$ corresponding to the connectivity vector $\mathbf{m}$, and the dynamics of $\mathbf{x}(t)$ is expressed as

$$\mathbf{x}(t) = \kappa(t)\mathbf{m}, \tag{103}$$

with $\kappa(t)$ given by

$$\kappa(t) = \frac{\mathbf{x}(t)^{\mathsf{T}}\mathbf{m}}{||\mathbf{m}||^2}. \tag{104}$$

Substituting Eq (103) into Eq (100) and inserting the unit-rank connectivity, the dynamics of the latent variable $\kappa$ can be expressed as

$$\dot{\kappa}(t) = -\kappa(t) + \kappa_{rec}(t) \tag{105}$$

where

$$\kappa_{rec}(t) = \frac{1}{N}\sum_{i=1}^{N}n_i\phi(\kappa(t)m_i).$$

(106)

The quantity $\kappa_{rec}(t)$ represents the total recurrent input to $\kappa$. The sum in the r. h. s. of Eq (106) can moreover be interpreted as the empirical average of $n_i\phi(\kappa(t)m_i)$ over the neurons in the network. In the limit of large network size $N$, this average converges to the integral of $n\phi(\kappa(t)m)$ over the distribution $P(m, n)$ of the components of connectivity vectors. For low-rank networks, the mean-field limit corresponds to replacing $\kappa_{rec}(t)$ with this integral [31, 33]:

$$\kappa_{rec} = \int \mathrm{d}m\mathrm{d}nP(m, n)n\phi(\kappa m).$$

(107)

In the Gaussian-mixture low-rank model, each neuron $i$ is assigned to a population $p$ for $p = 1\ldots P$. Within each population, the components $(m_i, n_i)$ are generated from a multivariate Gaussian distribution $P^p(m, n)$, that is

$$P^p(m, n) = \mathcal{N}\left(\begin{pmatrix}\bar{m}^p \\ \bar{n}^p\end{pmatrix}, \begin{pmatrix}\sigma_{m^p}^2 & \sigma_{nm}^p \\ \sigma_{nm}^p & \sigma_{n^p}^2\end{pmatrix}\right).$$

(108)

In the mean-field limit, $\kappa_{rec}$ is therefore given by

$$\kappa_{rec} = \sum_{p=1}^{P}\alpha_p\int \mathrm{d}m\mathrm{d}nP^p(m, n)n\phi(\kappa m),$$

(109)

where $\alpha_p$ is the fraction of neurons in population $p$.

Integrating by parts, $\kappa_{rec}$ can be re-expressed as (S1 Text)

$$\kappa_{rec} = \sum_{p=1}^{P}\alpha_p(\bar{n}^p\langle\phi(\mu_x^p, \Delta_x^p)\rangle + \langle\phi'(\mu_x^p, \Delta_x^p)\rangle\sigma_{nm}^p\kappa).$$

(110)

Here $\mu_x^p$, $\Delta_x^p$ are the mean and variance of the inputs to population $p$, given by

$$\begin{aligned}\mu_x^p &= \kappa\bar{m}^p, \\ \Delta_x^p &= \kappa^2\sigma_{m^p}^2,\end{aligned}$$

(111)

and the symbol $\langle f(\mu, \Delta)\rangle$ stands for the expected value of a function $f(x)$ with respect to a Gaussian variable $x$ with mean and variance $\mu, \Delta$, that is

$$\langle f(\mu, \Delta)\rangle = \int dx(2\pi)^{-1/2}\exp(-x^2/2)f(\mu + \sqrt{\Delta}x).$$

(112)

Altogether, using MFT for Gaussian-mixture low-rank networks gives the closed dynamics of the latent variable $\kappa$:

$$\dot{\kappa} = -\kappa + \sum_{p=1}^{P}\alpha_p(\bar{n}^p\langle\phi(\kappa\bar{m}^p, \kappa^2\sigma_{m^p}^2)\rangle + \langle\phi'(\kappa\bar{m}^p, \kappa^2\sigma_{m^p}^2)\rangle\sigma_{nm}^p\kappa).$$

(113)

In particular, the corresponding steady state is given by

$$\kappa = \sum_{p=1}^{P} \alpha_p (\bar{n}^p \langle \phi(\kappa\bar{m}^p, \kappa^2\sigma_{m^p}^2)\rangle + \langle \phi'(\kappa\bar{m}^p, \kappa^2\sigma_{m^p}^2)\rangle \sigma_{nm}^p \kappa). \tag{114}$$

Note that the first and second terms on the r. h. s. respectively correspond to the mean and covariance of the entries of the unit-rank connectivity vectors **m** and **n**.

**2.6.2 Approximate dynamics for locally-defined connectivity.** We next apply the MFT to unit-rank connectivity obtained as an approximation of locally-defined connectivity for the different considered cases.

*Independent connectivity.* We start from the network with independent connectivity, in which case the unit-rank connectivity vectors obtained by approximating locally-defined connectivity have no covariance, i. e. $\sigma_{nm}^p = 0$ (Methods Sec. 2.5).

The dynamical system for the latent variable $\kappa$ therefore contains only the mean term

$$\dot{\kappa} = -\kappa + \sum_{p=1}^{P} \alpha_p \bar{n}^p \langle \phi(\kappa\bar{m}^p, \kappa^2\sigma_{m^p}^2)\rangle. \tag{115}$$

For the Gaussian random model, inserting the expressions for $\bar{m}^p$ and $\bar{n}^p$ (Eqs (54)–(56)), the fixed point obeys

$$\kappa = J_E \langle \phi(\kappa, \kappa^2\sigma_{m^E}^2)\rangle - J_I \langle \phi(\kappa, \kappa^2\sigma_{m^I}^2)\rangle, \tag{116}$$

where the variance $\sigma_{m^p}^2$ of connectivity components $m_i$ is given by Eq (90).

For the sparse random model, we further consider Eqs (38), (54)–(56) to obtain $\bar{n}^p$ here, and the fixed point is

$$\kappa = cN_E A_E \langle \phi(\kappa, \kappa^2\sigma_{m^E}^2)\rangle + cN_I A_I \langle \phi(\kappa, \kappa^2\sigma_{m^I}^2)\rangle, \tag{117}$$

where $\sigma_{m^p}^2$ is obtained from Eq (91).

*Reciprocal motifs.* Correlations between reciprocal connections lead to non-zero covariance $\sigma_{nm}^p$ between the unit-rank connectivity vectors obtained by approximating locally-defined connectivity (Methods Sec. 2.3, Eq (72)). The dynamical system for the latent variable $\kappa$ therefore contains both the mean and covariance terms (Eq (114)).

For the Gaussian random model, combining Eqs (54)–(56), (90) and (97) the fixed point obeys

$$\kappa = \sum_{p=E,I} \alpha_p (\bar{n}^p \langle \phi(\kappa\bar{m}^p, \kappa^2\sigma_{m^p}^2)\rangle$$
$$+ \frac{1}{\lambda^2} \langle \phi'(\kappa\bar{m}^p, \kappa^2\sigma_{m^p}^2)\rangle (J_E g_{pE} g_{Ep} \eta_{Ep} - J_I g_{Ip} g_{pI} \eta_{Ip})\kappa) \tag{118}$$

with the variance $\sigma_{m^p}^2$ of connectivity components $m_i$ given by Eq (90). For the sparse model, combining Eqs (38), (54)–(56), (91) and (99), the fixed point obeys

$$\kappa = \sum_{p=E,I} \alpha_p (\bar{n}^p \langle \phi(\kappa\bar{m}^p, \kappa^2\sigma_{m^p}^2)\rangle$$
$$+ \frac{1}{\lambda^2} \langle \phi'(\kappa\bar{m}^p, \kappa^2\sigma_{m^p}^2)\rangle NA_p c^2 \left( A_E^2 N_E(\rho_{Ep} - c) + A_I^2 N_I(\rho_{Ip} - c)\right)\kappa). \tag{119}$$

with the variance $\sigma_{m^p}^2$ of connectivity components $m_i$ given by Eq (91).

**2.6.3 Mean-field theory for superpositions of low-rank and full rank random connectivity.**   Here we review an alternate form of mean-field theory for random connectivity consisting of a superposition of a low-rank structure and full-rank random part [30, 32]. This form of MFT can be directly applied to independently generated connections, where the connectivity matrix consists precisely of a superposition of a low-rank part corresponding to the mean, and a full-rank random part corresponding to fluctuations (Eqs (24) and (121)). Extending this type of MFT to the situation where reciprocal connections are present is however challenging [18]. Moreover, in contrast to the case where connectivity is exactly low-rank, when the additional full-rank random part is present the mean-field theory describes only the steady-state activity (and linearized dynamics around it), but not the full dynamics as in Eq (100).

The key assumption of MFT for randomly connected networks is that the total input $x_i$ to each unit can be approximated as a stochastic Gaussian process [53]. The first two cumulants (mean and variance) of that Gaussian process are then computed self-consistently to characterize the steady-state activity.

At a fixed point, the total input $x_i$ obeys

$$x_i = \sum_{j=1}^{N} J_{ij} \phi(x_j). \tag{120}$$

Replacing $J_{ij}$, where $i, j$ belong to populations $p, q$ respectively, by the superposition of rank-one mean and full-rank random connectivity components $\bar{m}^p \bar{n}^q / N + z_{ij}$ we get

$$x_i = \frac{\bar{m}^p}{N} \sum_{q=1\dots P} \bar{n}^q \sum_{j \in N_q} \phi(x_j(t)) + \sum_{j=1}^{N} z_{ij} \phi(x_j). \tag{121}$$

Denoting by $[\cdot]$ the average over the distribution of $x_i$, the mean of $x_i$ can then be expressed as

$$[x_i] = \bar{m}^p \bar{\kappa} \tag{122}$$

where we introduced

$$\bar{\kappa} = \sum_{p=1}^{P} \bar{n}^p \sum_{i \in N_p} [\phi(x_i)]/N \tag{123}$$

and we assumed that the zero-mean random connectivity $z_{ij}$ is uncorrelated with the firing rate $\phi(x_j)$, so that

$$\sum_{j=1}^{N} [z_{ij} \phi(x_j)] = 0. \tag{124}$$

Similarly, the correlation between $x_i$ and $x_j$, where $i \in N_p$ and $j \in N_q$, is given by

$$
\begin{aligned}
\left[x_i x_j\right] \quad &= \frac{\bar{m}^p}{N} \sum_{s=1}^{P} \bar{n}^s \sum_{k \in N_s} [\phi(x_k)] \frac{\bar{m}^q}{N} \sum_{t=1}^{P} \bar{n}^t \sum_{l \in N_t} [\phi(x_l)] \\
&+ \frac{\bar{m}^p}{N} \sum_{s=1}^{P} \bar{n}^s \sum_{k \in N_s} [\phi(x_k)] \sum_{l=1}^{N} [z_{jl} \phi(x_l)] \\
&+ \frac{\bar{m}^q}{N} \sum_{t=1}^{P} \bar{n}^t \sum_{l \in N_t} [\phi(x_l)] \sum_{k=1}^{N} [z_{ik} \phi(x_k)] \\
&+ [\sum_{l=1}^{N} z_{jl} \phi(x_l) \sum_{k=1}^{N} z_{ik} \phi(x_k)] \\
&= \bar{m}^p \bar{m}^q \bar{\kappa}^2 + \delta_{ij} \sum_{k=1}^{N} [z_{ik} z_{jk}][\phi^2(x_k)]
\end{aligned}
\tag{125}
$$

where we assume the neuronal activities are decorrelated $[\phi(x_i)\phi(x_j)] = [\phi(x_i)][\phi(x_j)]$ when $i \neq j$. This assumption holds for independently-generated connections, but not in presence of reciprocal correlations [18]. The covariance between $x_i$ and $x_j$ therefore becomes

$$
\left[x_i x_j\right] - [x_i][x_j] = 
\begin{cases}
\sum_{k=1}^{N} [z_{ik}^2][\phi^2(x_k)] & \text{for } i = j, \\
0 & \text{for } i \neq j.
\end{cases}
\tag{126}
$$

Within the mean-field approximation, neuronal activation $x_i$ are therefore uncorrelated Gaussian variables with mean and variance given by Eqs (122) and (126)

$$
\begin{aligned}
\mu_{x_i} &\coloneqq [x_i] = \bar{m}^p \bar{\kappa}, \\
\Delta_{x_i} &\coloneqq [x_i^2] - [x_i]^2 = \sum_k [z_{ik}^2][\phi^2(x_k)].
\end{aligned}
\tag{127}
$$

To determine $\bar{\kappa}$ and $[\phi(x_k)^2]$, we finally express Eqs (123) and (127) as Gaussian integrals over $x_i$ in population $p$:

$$
\begin{aligned}
\bar{\kappa} &= \sum_{q=1}^{P} \alpha_q \bar{n}^q \langle \phi(\mu_x^q, \Delta_x^q) \rangle, \\
\sum_{k=1}^{N} [z_{ik}^2][\phi(x_k)^2] &= \sum_{q=1}^{P} N_q \sigma_{z_{pq}}^2 \langle \phi^2(\mu_x^q, \Delta_x^q) \rangle.
\end{aligned}
\tag{128}
$$

Here we replaced $\bar{m}^p = 1$ and $\sum_{k=1}^{N} [z_{ik}^2] f(\cdot) = \sum_{q=1}^{P} N_q \sigma_{z_{pq}}^2 f(\cdot)$ given the eigenvector normalization in Eq (54), and the assumption that variances $[z_{ik}^2]$ depend on the populations the units $i$ and $k$ belong to (Eqs (26), (30) and (54)). Therefore, the stationary mean and variance

of the dynamics of synaptic inputs in population $p$ are

$$\mu_x^p \quad := \bar{\kappa},$$

$$\Delta_x^p \quad := \sum_{q=1}^{P} N_q \sigma_{z_{pq}}^2 \langle \phi^2(\mu_x^q, \Delta_x^q) \rangle. \tag{129}$$

Eqs (128) and (129) give the self-consistent equations for the stationary solutions of the dynamics.

More specifically, in the Gaussian random model, we combine connectivity statistics given by Eqs (26), (54)–(56), so that we have

$$\mu_x^p \quad := J_E \langle \phi(\mu_x^E, \Delta_x^E) \rangle - J_I \langle \phi(\mu_x^I, \Delta_x^I) \rangle,$$

$$\Delta_x^p \quad := \sum_{q=E,I} \alpha_q g_{pq}^2 \langle \phi^2(\mu_x^q, \Delta_x^q) \rangle, \tag{130}$$

while for the sparse random model, we combine connectivity statistics given by Eqs (29) and (30), (38), (54)–(54), so that we have

$$\mu_x^p \quad := c N_E A_E \langle \phi(\mu_x^E, \Delta_x^E) \rangle + c N_I A_I \langle \phi(\mu_x^I, \Delta_x^I) \rangle,$$

$$\Delta_x^p \quad := \sum_{q=E,I} N_q c(1-c) A_q^2 \langle \phi^2(\mu_x^q, \Delta_x^q) \rangle. \tag{131}$$

## Supporting information

**S1 Text. Dynamics in Gaussian-mixture low-rank networks.**
(PDF)

**S2 Text. Linear stability at fixed points in rank-one networks.**
(PDF)

**S3 Text. Comparison between dynamics in full-rank connectivity with rank-one approximation.**
(PDF)

**S4 Text. Chaotic dynamical transition point for networks with independent connectivity.**
(PDF)

**S5 Text. Local connectivity statistics in rank-$R$ Gaussian-mixture models.**
(PDF)

**S6 Text. First-order approximation of eigenvalues and eigenvectors.**
(PDF)

**S1 Fig. Comparison of singular value decomposition- (SVD-) and eigendecomposition-based low-rank approximation.** Blue scatters in (A, B) show eigenvalue spectra of the Gaussian excitatory-inhibitory full rank matrices $\mathbf{J}$, with in general rank-2 mean connectivity $\bar{\mathbf{J}}$, and i. i. d. random parts with identical variances $g^2/N$ over neurons. Blue dots in the circular bulk show $N-1$ complex eigenvalues for one realization of the random connectivity, outlying eigenvalues (blue dots) are shown for 30 realizations of the random connectivity. Dashed envelopes indicate the theoretical predictions for the radius $r_g = g$ of the circular bulk, red circles represent one of the eigenvalue of $\bar{\mathbf{J}}$ corresponding to the outlier of $\mathbf{J}$. Network parameters $N_E$

$= 2N_I = 600$, $N = N_E + N_I$, $g = 0.8$, $\bar{J}_{EE} = 0.0018$, $\bar{J}_{IE} = 0.0015$, $\bar{J}_{EI} = 0$, $\bar{J}_{II} = -0.0013$. Purple triangles in (A) show the eigenvalues of the eigendecomposition-based rank-one approximation for the corresponding 30 realizations of the full rank matrices. Their location on the $y$-axis is shifted to help visualization. Purple triangles in (B) show the eigenvalues of the SVD-based rank-one approximation for the corresponding 30 realizations of the full rank matrices. (TIF)

## Acknowledgments

We are grateful to David Dahmen, Eric Shea-Brown and Stefano Recanatesi for helpful discussions during the revision of this manuscript.

## Author Contributions

**Conceptualization:** Yuxiu Shao, Srdjan Ostojic.

**Data curation:** Yuxiu Shao.

**Formal analysis:** Yuxiu Shao.

**Funding acquisition:** Srdjan Ostojic.

**Investigation:** Yuxiu Shao.

**Methodology:** Yuxiu Shao, Srdjan Ostojic.

**Project administration:** Srdjan Ostojic.

**Resources:** Srdjan Ostojic.

**Software:** Yuxiu Shao.

**Supervision:** Srdjan Ostojic.

**Validation:** Yuxiu Shao, Srdjan Ostojic.

**Visualization:** Yuxiu Shao.

**Writing – original draft:** Yuxiu Shao.

**Writing – review & editing:** Yuxiu Shao, Srdjan Ostojic.

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
