## [Decision Letter · Decision Letter 0]

15 Oct 2022

Dear Shao,

Thank you very much for submitting your manuscript "Relating local connectivity and global dynamics in recurrent excitatory-inhibitory networks" for consideration at PLOS Computational Biology.

As with all papers reviewed by the journal, your manuscript was reviewed by members of the editorial board and by several independent reviewers. In light of the reviews (below this email), we would like to invite the resubmission of a significantly-revised version that takes into account the reviewers' comments.

We cannot make any decision about publication until we have seen the revised manuscript and your response to the reviewers' comments. Your revised manuscript is also likely to be sent to reviewers for further evaluation.

Sincerely,

Michele Migliore

Academic Editor

PLOS Computational Biology

Marieke van Vugt

Section Editor

PLOS Computational Biology

Reviewer's Responses to Questions

**Comments to the Authors:**

Reviewer #1: Classic models in theoretical neuroscience rely on unstructured connectivity: either Erdos-Renyi (or block-ER) or independent Gaussian weights. Two lines of result have called these models into question. First, physiological and anatomical studies have shown that patterns of connectivity between neurons (motifs) are over-represented compared to their chance levels in Erdos-Renyi networks. Second, the division of neurons into discrete classes suggests a low-rank contribution to the connectivity. In addition, trained neural network models often exhibit a learned low-rank connectivity. Here, Shao and Ostojic relate the dynamics of these two classes of networks.

This is a good paper, with work of the high quality I’ve come to expect in PLOS Computational Biology. Its organization and presentation, however, need some improvement. I also have some clarifying questions about several points.

First, regarding organization: The text surrounding Figures 3 and 4 does not at all match the organization of the figures. For example, starting in section 1.4, we have the first reference to figure 3: 3A. It is followed by references to Figures 4F-H, then 3M. This suggests that either the figures or the text should be significantly reorganized. Having two separate organizations for the figures and text is confusing to the reader. (Related to this point, I think Figures 3 and 4 could also easily be separated into multiple figures.) As it stands, the reader has to hunt around for the relevant figure panel, both skipping forwards/backwards to the relevant figure, and then searching within the figure. Once at a figure, it is natural to look a bit ahead/behind in the panels - but the divorce between figure and text organization means that the next panels I look at have nothing to do with the nearby text that sent me to the figure, leaving me unnecessarily confused and wondering why these panels are in the same figure. I leave it to the authors to decide which to revise (figures or text) but please choose a consistent presentation order for this section. The relation between Figures 5-9 and the text is much more clear and was a relief.

Second, regarding motifs: Throughout, the authors study networks with an over-representation of reciprocal connectivity motifs. What about other two-synapse motifs? In terms of the linearized dynamics, expanding (I-J)^{-1} in powers of J yields an expansion where the reciprocal motifs are at the same order as convergent, divergent, and feedforward chain motifs, and those other motifs also impact the statistics of fluctuations (Hu et al. 2013 J. Stat. Mech, 2014 PRE, 2018 PRE, Recanatesi PCB 2019, Dahmen et al. bioRxiv 2020). Can networks with an over-representation of these other two-synapse motifs also be mapped on to a low rank Gaussian mixture network?

Third: Regarding the sparse network. In Eq. 36 for example, how do higher-order statistics of the synaptic weights in the sparse network scale with N? If p is O(1/N) but the non-zero connections have O(1) strength, I think the higher cumulants are all of O(1/N) also. These are neglected in the Gaussian model, so we’d expect the Gaussian model to fail for stronger weights similarly to the classic Gaussian mean field theory for spiking networks (Ostojic 2014). Is this the case? How does this affect the relation to a low rank Gaussian mixture network?

Fourth, regarding terminology. Throughout, the authors refer to the left and right eigenvectors of an arbitrary matrix J. For J to have an eigendecomposition it must be diagonalizable. Is this guaranteed for the networks studied? And if so, aren’t the left and right eigenvectors equal? Or, are these the left and right singular vectors of the connectivity? If so, I’d suggest using that terminology for accessibility to a broader readership.

Fifth, regarding matrix perturbation theory. Perturbations usually refer to a small parameter. Are there requirements on the norm of Z? Truncating in Eq. 6 seems to imply a small norm of Z, although this isn’t stated explicitly. How does this relate to the norm of J, and to the mapping from an excitatory-inhibitory or sparse network? I’m not sure that I would expect the random part to be weaker than the structured part in these cases.

Minor points:

Figure 2E: Where are the red curves for \\sigma^2_{m^E}, \\sigma^2_{n^E}?

Line 183/184: [\\theta_k] = 0, for all k greater than 0? theta_0 is independent of Z, I think.

Eq. 17: What is \\alpha_p? I can’t seem to find a definition for it in the surrounding text. Is \\sigma^p_{nm} = \\sigma_{n^p m^p}?

Line 319: When only the first term is present int he res of Eq. 17, it reduces to the classic Amari-Grossberg rate model, which seems worth mentioning.

Figure 5C: blue and purple is a low-contrast color choice that makes this harder to read.

Figures 5 and 6D: the dark colors here were also hard to quickly distinguish.

Figures 5 and 6: How does the bistability in \\kappa manifest in activity of the underlying network?

Eq. 20: It looks like J should be regular typeface on the right-hand side.

Eq. 29: If square brackets denote the average over the distribution of J, what does ()_J denote?

Reviewer #2: The review is uploaded as an attachment.

Reviewer #3: The main contributions of the manuscript are (1) deriving a rank-1 approximation to recurrent connectivity defined by local connection statistics and reciprocal motifs and (2) using the rank-1 approximation to predict the dynamics in E-I networks. These results are very interesting and original, both conceptually and on a technical level. The manuscript is mostly very well written. My main concerns and suggestions are on the interpretation and discussion of the literature, as detailed below.

Major:

1. When describing single neuron results such as Fig 2D, the authors referred to them as “predictions of the theory.” But this is potentially misleading because to obtain the eigenvector entries of the prediction, one needs to know the matrix Z. This usually means knowing the connectivity matrix J, under which case one can compute the exact eigenvector without using the perturbation theory. Note that the situation is very different for the results on the statistical level, such as Fig 2B, E. These can be directly calculated from the *local statistics* of J, making them true predictions from the theory. Similar issues also occur in the dynamic results, such as Fig. 6D. I suggest carefully distinguishing these two types of results throughout the manuscript including the abstract, and perhaps rephrasing the single neuron results other than “predictions.”

2. The conditions and limitations of the method should be more clearly discussed. For example, the rank-1 approximation of connectivity will not work when lambda lies within the bulk spectrum (line 410 should be moved/repeated upfront). The case when lambda is within the bulk is related to balanced EI input on average and has important implications on the dynamics (e.g., ref 44) and thus is worth discussing. For predicting dynamics with rank-1 connectivity, I expect the approach would not apply when the leading eigenvalue is negative (e.g., Fig. 4F), even if the approximation of the connectivity is accurate. It should also be made clear in the abstract that the study specifically focused on the reciprocal motifs rather than general types of motifs.

3. Related to point 2, the claims about the generality of the approach, unless directly demonstrated, should be refrained from being made. For example, in the abstract, “can in general be approximated by…”, line 503 in Discussion “can be directly extended to networks with additional structure,” line 519 “can be extended to various forms of local connectivity motifs,” and line 520 “can be applied more broadly to study experimentally-obtained …” I suggest reducing or rephrasing them as directions for future work and potential impact of the current study.

4. When describing the numerical results/figures, it should be noted which part of the numerics corresponds to finite-size effects vs. systematic errors due to the approximations. It will also be helpful to explain (more prominently) the asymptotic properties, i.e., the orders of the variables such as m, n, theta_k as N-> inf.

5. The second part of the paper shows that the dynamics of a recurrent neural network can be approximated by only considering the leading eigenvector and ignoring its bulk spectrum. I find this result by itself very interesting. But a discussion is missing on whether this question or closely related ones has been studied in the literature, particularly the previous work by the authors.

6. Regarding literature: The first part on the perturbation eigenvalues is closely related to this paper on random matrix, which should be cited and discussed, e.g., in line 485: Tao 2013, Outliers in the spectrum of iid matrices with bounded rank perturbations. The relation between second-order motifs and the low rank component of connectivity and outliers in the covariance spectrum has been studied in ref 62. This should be discussed, e.g., in line 484, 505. The biostability dynamics due to local statistics and motifs is conceptually related to Nykamp et al. PRE 2017 Mean-field equations for neuronal networks with arbitrary degree distributions, which should be cited.

Minor:

The Jbar in Eq. 3 is also a naive low rank approximation to J by discarding the noise part Z. Many results could benefit from a discussion using Jbar as a benchmark. This has been done in the dynamics part, such as line 304, 328. But the comparison may be worth adopting explicitly throughout the manuscript.

The difference in the leading eigenvalue perturbation for the iid Z case vs. reciprocal motif case should be emphasized (e.g., in the Discussion). For example, the truncation Eq.7 is, in fact, exact as N->inf for iid Z but is strictly an approximation with reciprocal motifs.

The rationale of truncating the expansion eq.6 at third order should be stated (it is the first nontrivial term for the reciprocal case).

Is it possible to have more than three outlier eigenvalues for the reciprocal motif case? It seems to be the case considering keeping higher order terms in Eq. 6.

The complex eigenvalues of an asymmetric matrix such as J lack an intrinsic ordering. The author uses the absolute value to define the leading eigenvalue. But I am not sure this should always be the case. For example, when there are fewer reciprocal motifs, i.e., negative eta (similar to Fig. 9B), the elliptical bulk spectrum could exceed the magnitude of outlier lambda.

Please consider adding a reference to the equations when referring to terms or methods such as the “locally defined connectivity” to remind the reader (e.g., line 337).

It will be helpful to describe briefly the organization and logic of the sections (e.g., Gaussian connectivity first, then generalize to sparse connectivity) at the end of the introduction.

Eq. 4 should be referenced again between Eq. 5 and Eq. 6.

In Fig. 3B, H, N, why do the theory curves for lambda_c1 and lambda_c2 not have exactly the same real part? They are complex conjugates from the third order polynomial, right?

In Fig. 4 caption (E), k is not defined/referenced.

It is worth mentioning in the summary of section 1.3 (its last paragraph) that for iid Z, lambda=lambda0.

A brief derivation on how to apply ref 47 to get Eq. 9 would be helpful.

Line 122, higher rank approximations?

Line 360, “the major novel effect,” and other comparisons between iid and reciprocal motif cases should be discussed and summarized again in the Discussion for better visibility.

Line 520, the ref 10 -> ref 22?

**Have the authors made all data and (if applicable) computational code underlying the findings in their manuscript fully available?**

Reviewer #1: None

Reviewer #2: Yes

Reviewer #3: **No: **The github link given to the code in the does not work for me.

PLOS authors have the option to publish the peer review history of their article (what does this mean?). If published, this will include your full peer review and any attached files.

Reviewer #1: No

Reviewer #2: **Yes: **Laureline Logiaco

Reviewer #3: No
---

## [Decision Letter · Decision Letter 1]

6 Jan 2023

Dear Shao,

We are pleased to inform you that your manuscript 'Relating local connectivity and global dynamics in recurrent excitatory-inhibitory networks' has been provisionally accepted for publication in PLOS Computational Biology.

Best regards,

Michele Migliore

Academic Editor

PLOS Computational Biology

Marieke van Vugt

Section Editor

PLOS Computational Biology

Reviewer's Responses to Questions

**Comments to the Authors:**

Reviewer #1: The authors have thoroughly addressed my previous comments and questions.

Reviewer #2: My requests have been addressed. I commend the authors for the new S6 appendix, which is extremely helpful. The manuscript is ready for publication.

PS: I believe that there remains a small typo on line 701 - I think that a parenthesis in front of the factor 1/2 is missing, such that the authors meant gamma_pq = sqrt( ( 1-sqrt(1-eta**2) ) * (1/2) ) or gamma_pq = - sqrt( ( 1-sqrt(1-eta**2) ) * (1/2) ).

Reviewer #3: The authors’ careful revision and responses have sufficiently addressed all my concerns, and I wholeheartedly recommend publication.

Just to clarify on “major comment 6”, ref 62 studied the covariance eigenvalue outlier due to diverging motifs (Fig. 5BDEF, caption) but not the impact of reciprocal motifs on these outliers.

Please also check the code link/public accessibility, which does not yet work for me.

**Have the authors made all data and (if applicable) computational code underlying the findings in their manuscript fully available?**

Reviewer #1: None

Reviewer #2: Yes

Reviewer #3: Yes

PLOS authors have the option to publish the peer review history of their article (what does this mean?). If published, this will include your full peer review and any attached files.

Reviewer #1: No

Reviewer #2: **Yes: **Laureline Logiaco

Reviewer #3: No

---

## [Editor Report · Acceptance letter]

17 Jan 2023

PCOMPBIOL-D-22-01289R1 

Relating local connectivity and global dynamics in recurrent excitatory-inhibitory networks

Dear Dr Shao,

I am pleased to inform you that your manuscript has been formally accepted for publication in PLOS Computational Biology. Your manuscript is now with our production department and you will be notified of the publication date in due course.

With kind regards,

Timea Kemeri-Szekernyes
